# Broad-scale factors shaping the ecological niche and geographic distribution of *Spirodela polyrhiza*

**Marlon E. Cobos** *, A. Townsend Peterson

Department of Ecology and Evolutionary Biology & Biodiversity Institute, University of Kansas, Lawrence, Kansas, United States of America

* manubio13@gmail.com

**Data Availability Statement:** All the data used in this project can be obtained using the code provided or downloaded from the source websites described in the Methods section. Filtered occurrence data and R code to reproduce all

## Abstract

The choice of appropriate independent variables to create models characterizing ecological niches of species is of critical importance in distributional ecology. This set of dimensions in which a niche is defined can inform about what factors limit the distributional potential of a species. We used a multistep approach to select relevant variables for modeling the ecological niche of the aquatic *Spirodela polyrhiza*, taking into account variability arising from using distinct algorithms, calibration areas, and spatial resolutions of variables. We found that, even after an initial selection of meaningful variables, the final set of variables selected based on statistical inference varied considerably depending on the combination of algorithm, calibration area, and spatial resolution used. However, variables representing extreme temperatures and dry periods were more consistently selected than others, despite the treatment used, highlighting their importance in shaping the distribution of this species. Other variables related to seasonality of solar radiation, summer solar radiation, and some soil proxies of nutrients in water, were selected commonly but not as frequently as the ones mentioned above. We suggest that these later variables are also important to understanding the distributional potential of the species, but that their effects may be less pronounced at the scale at which they are represented for the needs of this type of modeling. Our results suggest that an informed definition of an initial set of variables, a series of statistical steps for filtering and exploring these predictors, and model selection exercises that consider multiple sets of predictors, can improve determination of variables that shape the niche and distribution of the species, despite differences derived from factors related to data or modeling algorithms.

## Introduction

Ecological niche modeling (ENM) includes a diverse set of tools used to study potential distributions of species via characterizations of their environmental requirements [1,2]. In particular, correlational ENMs use distributional data (occurrence records) and sets of environmental variables to calibrate models that are used to predict environmental suitability for a species

analyses is available from a GitHub repository (www.github.com/marlonecobos/Spirodelap_ENM).

**Funding:** This research was supported in part by a grant from the National Science Foundation (OIA-1920946). The funders had no role in study design, data collection and analysis, decision to publish, or preparation of the manuscript.

**Competing interests:** The authors have declared that no competing interests exist.

across a region of interest [3]. Variables appropriate to characterizing and understanding a species' niche are those that allow identifying conditions that are favorable for the species, as well as detecting potential limits of what is or is not suitable for a species (e.g., temperatures that allow maximum growth rates, or maximum temperatures that can be tolerated). However, differences in the scale at which ecological processes occur and the grain and extent to which environmental variables are measured make it difficult to select predictors based on direct interpretations of their biological importance [4].

The challenge of selecting appropriate environmental variables when characterizing species' ecological niches using correlative models is well-known in the field of distributional ecology [5–7]. In general, models can be constructed with two main goals: (1) to improve predictions of the geographic distribution of the species, and (2) to understand which environmental variables are important constraints on species' niches. When the goal is to improve model predictive ability, variables can be selected based on how they improve predictions of independent testing data. In these cases, environmental data sets that efficiently summarize environmental variation across an area of interest (e.g., principal components) are commonly considered to be good choices [8]. However, when models are constructed to understand which and how environmental variables shape species' ecological niches and geographic distributions, use of biologically meaningful and interpretable variables becomes more relevant [4].

Common procedures for selecting environmental predictors in ENM include reducing multicollinearity, testing contribution of variables to models, selecting variables based on their biological importance considering empirical evidence or the experience of researchers, or using a broad set of variables and letting the algorithm select important variables [9]. Other alternatives include selecting predictors by transforming original variables to summarize the variance explained by a set of principal components that are more information-rich and in general, are not correlated [10,11]. However, interpretation of the role of particular environmental factors in the characterization of species niches is complicated. More recently, different sets of variables have been used as part of the parameterizations to be tested during the process of model calibration [12]. After model selection, one or more sets of variables can be identified as more appropriate and powerful for use in creating final models.

The reality is that, regardless of the method used to determine the set of variables for modeling ecological niches, the decision is always difficult and the answer is rarely unique or unambiguous [9]. This complication exists because every step taken to define which variables are best may result in distinct sets of predictors at the end. For instance, when selecting one variable depending on correlation values and biological importance, the decision of which variable to keep depends on the researcher; many times, such an initial decision determines which other variables can or cannot be considered. Analyses like the variance inflation factor may end up identifying unique sets of variables; however, the set of variables selected depends on a predefined limit, which is not a biological consideration [13]. Distinct answers are obtained when different sets of variables are considered in model calibration, although such sets are usually subject to *a priori* processes of selection [14].

Implications of using one set or another set of environmental dimensions when creating models are not negligible, especially in applications in which model transfers to other geographic or temporal scenarios are required [15–17]. Other less explored complications are the effects that areas for model calibration, spatial resolution of raster layers used as predictors, and use of different modeling algorithms have on the variables that get selected. The area across which a model is calibrated has direct implications on predictions: models may be over- or under-fitted if such an area is poorly defined [18,19]. Little is known about how changes in calibration areas affect decisions related to variable selection during the modeling process. The spatial resolution of variables is known to affect model calibration and model transfers [6,20];

however, little has been said about its effects on the final set of variables selected (but see [21]). Distinct modeling algorithms may also perform better or worse depending on the sets of variables used, as all predictors influence the model and interact with other variables differently. Again, however, this factor has not been considered deeply (but see [22]), and the set of variables for modeling is usually fixed when using multiple algorithms (e.g., [23,24]).

Here we explore the challenges in defining sets of environmental variables in ENM for *Spirodela polyrhiza* (L.) Schleid (greater duckweed), a freshwater plant species with a broad near cosmopolitan distribution [25]. Specifically, we used distinct methods for variable selection in a multi-step approach. We performed analyses at two spatial resolutions, used two algorithms for model calibration, and considered different options regarding calibration areas, to explore the consequences of these factors on variable selection. We hypothesize that variables representing extreme conditions and environmental conditions during the active period of the species (see section Study organism) are better predictors for broad-scale characterizations of the species ecological niche and distribution. As little is known about macroecological factors driving the geographic distribution of greater duckweed, our explorations of environmental variables can help to understand the distributional potential of this plant, environmental dimensions limiting its potential for expansion to other areas, and how climate change might affect this species' range.

## Methods

### Study organism

*Spirodela polyrhiza* is a species of duckweed that ranks among the smallest angiosperms known (sizes 0.5–18 mm). It is a free-floating aquatic plant that reproduces vegetatively in largest part [26]. To overcome unfavorable conditions (specially during the winter), this species produces a starch-rich tissue called a turion that is denser than normal fronds, and sinks to the bottom of water sources until conditions become favorable [27]. Similar to other duckweed species, under appropriate conditions, *S. polyrhiza* grows at high rates, which helps it to cover large portions of the surface of the water bodies where it is present [28].

Considering their high growth rates, small size, simple structures, and potential for industrial applications, duckweed species have been the subject of intensive and detailed research [29]. Among the most notable applications are possible utility in water treatment [30], bioenergy [31], animal feeding [27], human nutrition [32,33], and pharmaceutical applications [34]. Given the potential of duckweed species as model organisms [35,36], stock collections of these species have been established by several institutions around the world, which have aided substantially in promoting further research on these species [37]. As such, various aspects of *S. polyrhiza* physiology, genome, and potential industrial applications have been studied in detail [27,38–41]. A remarkable characteristic of the geographic distribution of *S. polyrhiza* is that it extends worldwide. According to Les et al. [42], duckweed species and other aquatic plants dispersed transoceanically in the recent past, which highlights the importance of external biotic dispersal for this species [43]. Duckweed dispersal is mainly via adhesion to aquatic birds and mammals [25,44,45]. However, little has been done to understand the macroecological factors that drive its distribution, and only general aspects of the regions occupied by *S. polyrhiza* have been characterized [29].

### Occurrence data

We obtained geographic occurrence records for *S. polyrhiza* from the Global Biodiversity Information Facility (GBIF [46]) and the Botanical Information and Ecology Network (BIEN [47]). In all, a total of 85,923 georeferenced records were obtained (GBIF: 84,992; BIEN: 931).

We cleaned data from each database independently to exclude records from before 1970, lacking coordinates, with zero values for longitude and latitude, or duplicates [48]. Records marked as absent or uncommon were also removed from the GBIF data. After this initial cleaning, we had 45,913 records (GBIF: 45,459; BIEN: 454). We combined records from the two sources and excluded duplicates again. Records that were outside of, but closer than ~5' to the edge of environmental layers (i.e., that fell very close to informative areas for climate data) were moved to the nearest pixel with information; points falling farther outside layer borders were removed. To reduce bias from spatial autocorrelation, we thinned records using a minimum point-to-point distance of ~30'. We selected this value after testing the effect of increasing distances in the Moran's *I* statistic for all environmental variables (see S1 and S2 Tables). The final number of records after these procedures was 964. Occurrence data download, cleaning, and thinning were accomplished using rgbif [49], spooc [50], BIEN [51], and ellipsenm [52] in R [53].

## Environmental variables

We obtained raster environmental data layers from three sources: (1) bioclimatic (BIO) and solar radiation (SR) layers from WorldClim v2.1, at 10' resolution (available at www.worldclim.com [54]); (2) cation exchange capacity (CEC), organic carbon (OC), and pH, from the ISRIC–World Soil Information database, at 250 m resolution (available at www.soilgrids.org [55]); and (3) total phosphorus (TP), labile inorganic phosphorus (LIP), and organic phosphorus (OP) in soils, from Global Gridded Soil Phosphorus at 30' resolution (available at www.daac.ornl.gov/SOILS/guides/Global_Phosphorus_Dist_Map.html [56]). We used bioclimatic variables to represent temperature (which could help to identify thermal limits), and precipitation (which can inform about water availability). Solar radiation layers provide information on solar energy levels across a region in our models; soil variables offer more indirect information relevant to nutrient availability. All of these variables have been proven to be relevant to the development of the study species in analyses on local extents and/or in laboratory experiments [25,26] (S3 Table).

Solar radiation layers were available as averages for the 12 months of the year. To create layers that better represented extremes and annual averages, we created the following "bioclimatic-like" layers: annual mean solar radiation (AMSR), maximum solar radiation of the month with maximum values (SRMax), minimum solar radiation of the month with minimum values (SRMin), range of solar radiation (RSR), average solar radiation of the quarter with highest values (ASRQH), and average solar radiation of the quarter with lowest values (ASRQL). We created these variables using the values for the 12 months obtained from WorldClim. Variable processing and calculations were done using the packages raster [57] and gdalUtilities [58], in R.

To test the effect of spatial resolution on the outcome of variable selection processes, we created two groups of variables, at distinct spatial resolutions: (1) a group at 10' resolution including BIO and SR variables, plus CEC, OC, and pH; and (2) a group at 30' resolution including BIO and SR variables, plus TP, LIP, and OP. We performed raster aggregation procedures (average of values) on CEC, OC, and pH to match the resolution of BIO variables, and on BIO and SR variables to match the resolution of variables at 30'. One of the layers at 10' and at 30' resolution was used as a reference for the aggregation process to exactly match pixels among all layers at each resolution. The method of aggregation used was the nearest neighbor and the average value was used to represent environments aggregated. Although the set of variables representing soil conditions used at 10' differs from the one at 30', variable selection analyses

will help to identify whether the variables present in the two sets, at distinct resolutions, are consistently selected.

## Geographic areas for model calibration

To explore the effect of the areas across which models are calibrated on the set of variables selected, we explored four options for such areas in our analysis: (1) buffers of ~5˚ (~500 km at the Equator) around occurrence records, (2) concave-hull polygons with a buffer of ~5˚, (3) ecoregions occupied by the species buffered by ~1˚ (~100 km), and (4) the result of intersecting the previous three areas. Buffer distances for the first two types of calibration areas were defined considering that the species can be dispersed by birds over relatively large distances. Distance for ecoregion buffering was selected to include a more diverse set of environments around occupied regions. We obtained the layer of world terrestrial ecoregions from the Harvard WorldMap database (available at https://worldmap.maps.arcgis.com). Although a new simulation-based approach has been recently suggested as a reliable tool to estimate calibration areas [59], the broad distribution of this species makes it difficult for that method to be applied. The types of calibration areas used in our study have been used in other studies [9] to define relevant environments for model development (e.g., [59,60]). Our chosen calibration areas are therefore reasonable options to calibrate models considering that such areas should reflect what regions could have been accessible to the species and present relevant environments for comparisons (Fig 1). The two groups of variables were masked to the four areas. We created these calibration areas using the packages ellipsenm, rgeos [61], and rgdal [62] in R.

## Modeling algorithms

We used generalized linear models (GLMs) and Maxent [63,64] to estimate the ecological niche of the species. These algorithms are both used commonly in the literature and produce reliable and good-performing models [65,66]. For contrasts in model calibration, Maxent uses presences and a characterization of the background, whereas GLMs use presences and pseudo-absences; both background and the pseudo-absences were taken as a sample of available pixels across the calibration area. For purposes of comparison, the same points (20,000) were used in both algorithms, and were treated as both background and pseudo-absence data. The sample of 20,000 points was taken for each calibration area option independently. This number of points was used to achieve a good representation of the areas and corresponding environments over which presences will be compared, and to follow recommendations regarding amount of pseudo-absence data in ENM applications using regression techniques [67]. GLMs were performed as logistic models with a weight of 1 for presences and 10,000 for pseudo-absences (e.g., [14]). GLMs created in such a way are considered to be similar mathematically to Maxent models under certain conditions and assumptions [68].

## Variable selection process

As variable selection can be done in multiple ways and at distinct points in the process of data preparation or modeling, we followed a multi-step approach that considers quantitative and qualitative characteristics of predictors (Fig 2). Our approach consisted of (1) initial inspection and processing of variables (see section Environmental variables); (2) assessing linear correlations among variables; (3) exploring variable values in occurrences and across calibration areas; (4) an initial selection based on the criteria (2) and (3) and the biological relevance of variables; (5) creating variable sets resulting from all combinations of two or more initially selected variables; and (6) including all sets of variables in model calibration exercises to identify which algorithm parameters and variable sets, in concert, result in the best-performing

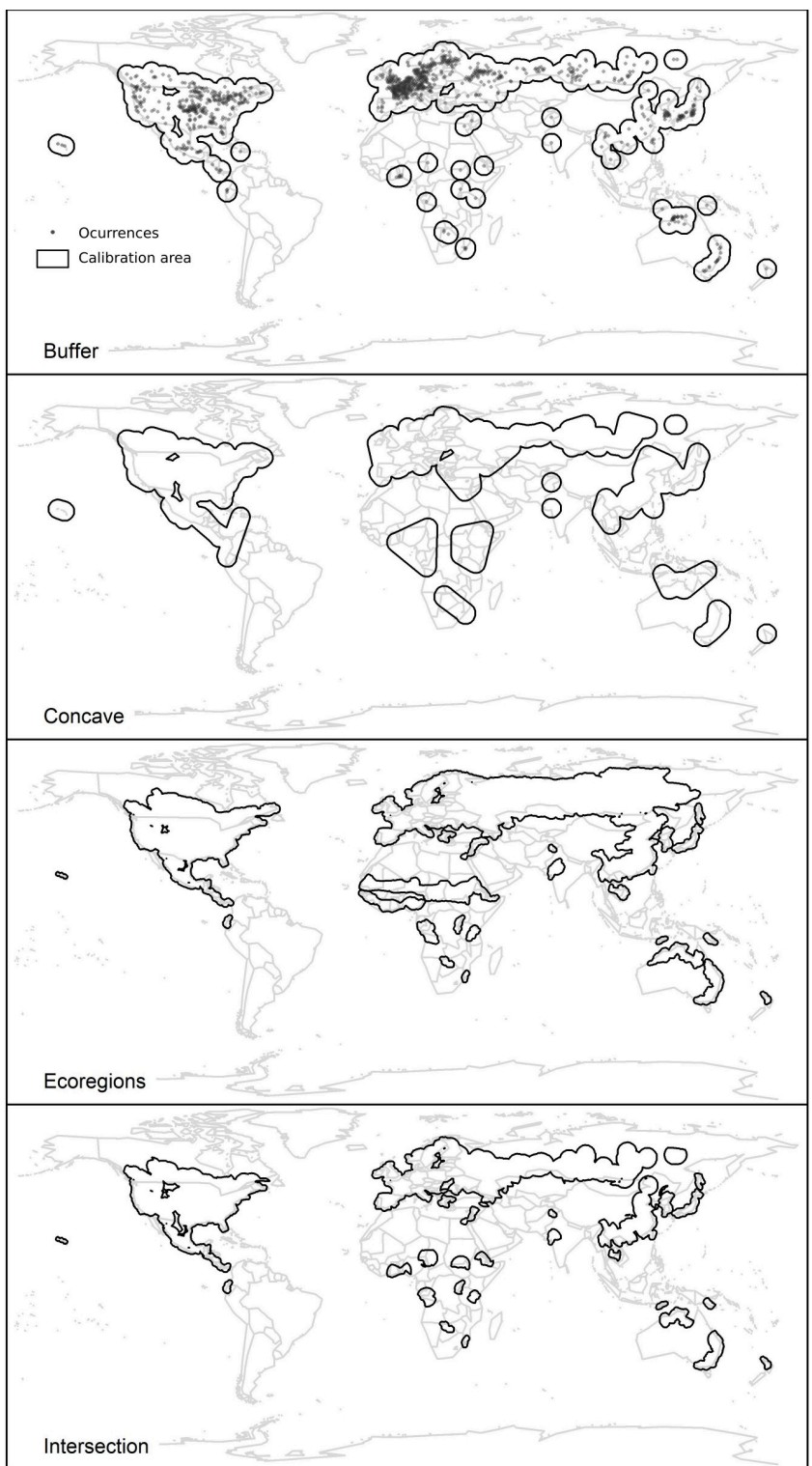

**Fig 1. Geographic representation of species occurrence data for *Spirodela polyrhiza* (upper panel) and areas for model calibration used to create ecological niche models.** Occurrence data represented are after filtering and spatial-thinning procedures. Buffer and concave areas are presented before masking them to land areas for purposes of representation.

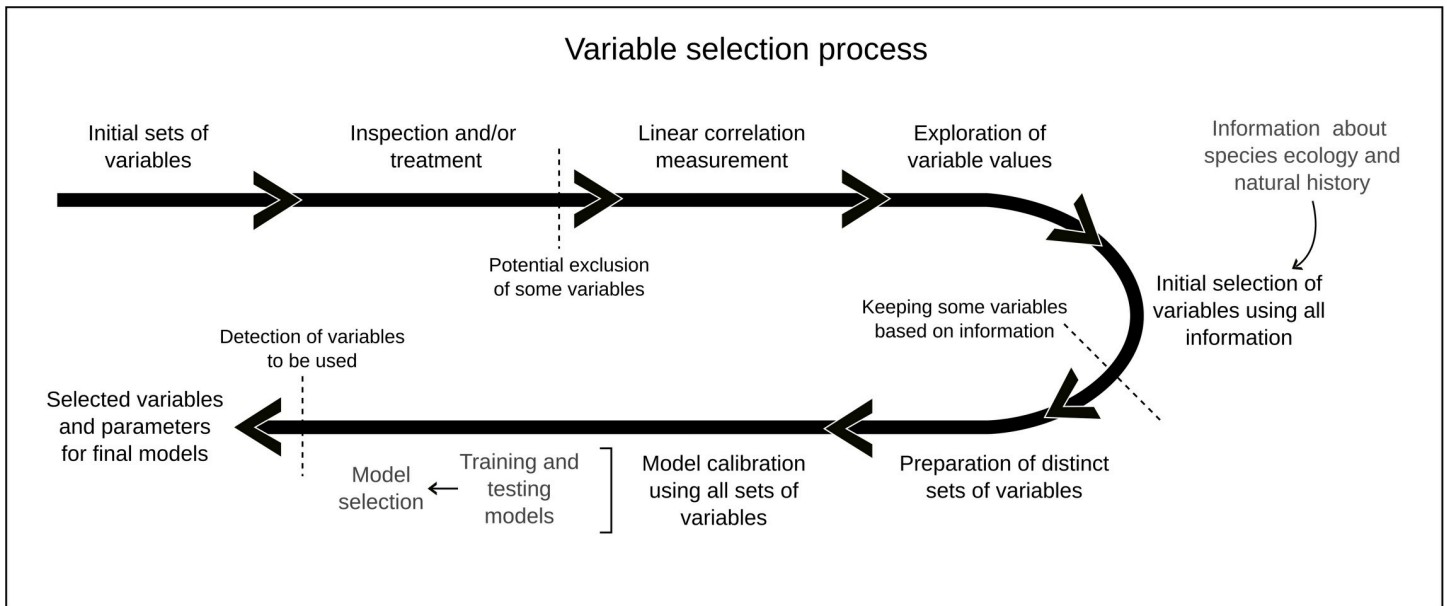

**Fig 2. Schematic representation of the process that was followed to select variables to model the ecological niche of the greater duckweed (*Spirodela polyrhiza*).**

models. We performed exploration exercises 16 times: the combination of two environmental data sets, four calibration areas, and two algorithms.

After the first two steps, initial selection of variables was done based on three considerations: (1) groups of variables with all variable-to-variable pairwise correlations $|r| \leq 0.8$ (Fig 3); (2) biological relevance of variables; and (3) variables for which the calibration area had wider limits in environmental dimensions than the occurrences [69], based on histogram plots of values (Fig 4). The latter consideration assumes that using variables for which the entire spectrum of responses can be characterized (i.e., non-truncated responses [2,70,71]) makes for better models [72]. Biological relevance of variables was determined based on details about the species' natural history [25,41], phenology [28,73], and physiology [38,74] in the literature, and our own experience with populations in the field and controlled environments. For simplicity, we selected the same initial set of variables based on the relevance criterion for the four calibration areas considered.

Using the groups of variables remaining after the initial selection, we prepared subsets of variables representing all combinations of three or more variables [14]. Such sets of variables were then used as part of our process of model calibration in which other parameter settings were tested. For both Maxent and GLMs, we tested five response types (lq, lp, q, qp, lqp; l = linear, q = quadratic, and p = product). For Maxent, six regularization multiplier values were explored (0.1, 0.3, 0.6, 1.0, 2.5, 5.0). Performance of candidate models was evaluated based on three criteria: statistical significance of predictions (partial ROC; [75]), omission rate (allowing a 5% omission error; [76]), and model fit and complexity (based on the Akaike information criterion for GLMs, and the AICc proposed by Warren and Seifert [77] for Maxent).

In total, then, for each model calibration exercise, 10,180 and 5065 GLM models were tested at 10' and 30', respectively, and 61,080 and 30,390 Maxent models were tested at 10' and 30', respectively. Model calibration processes with Maxent were done using the kuenm R package [12], and model calibration using GLMs was done using stats and other base functions in R [53].

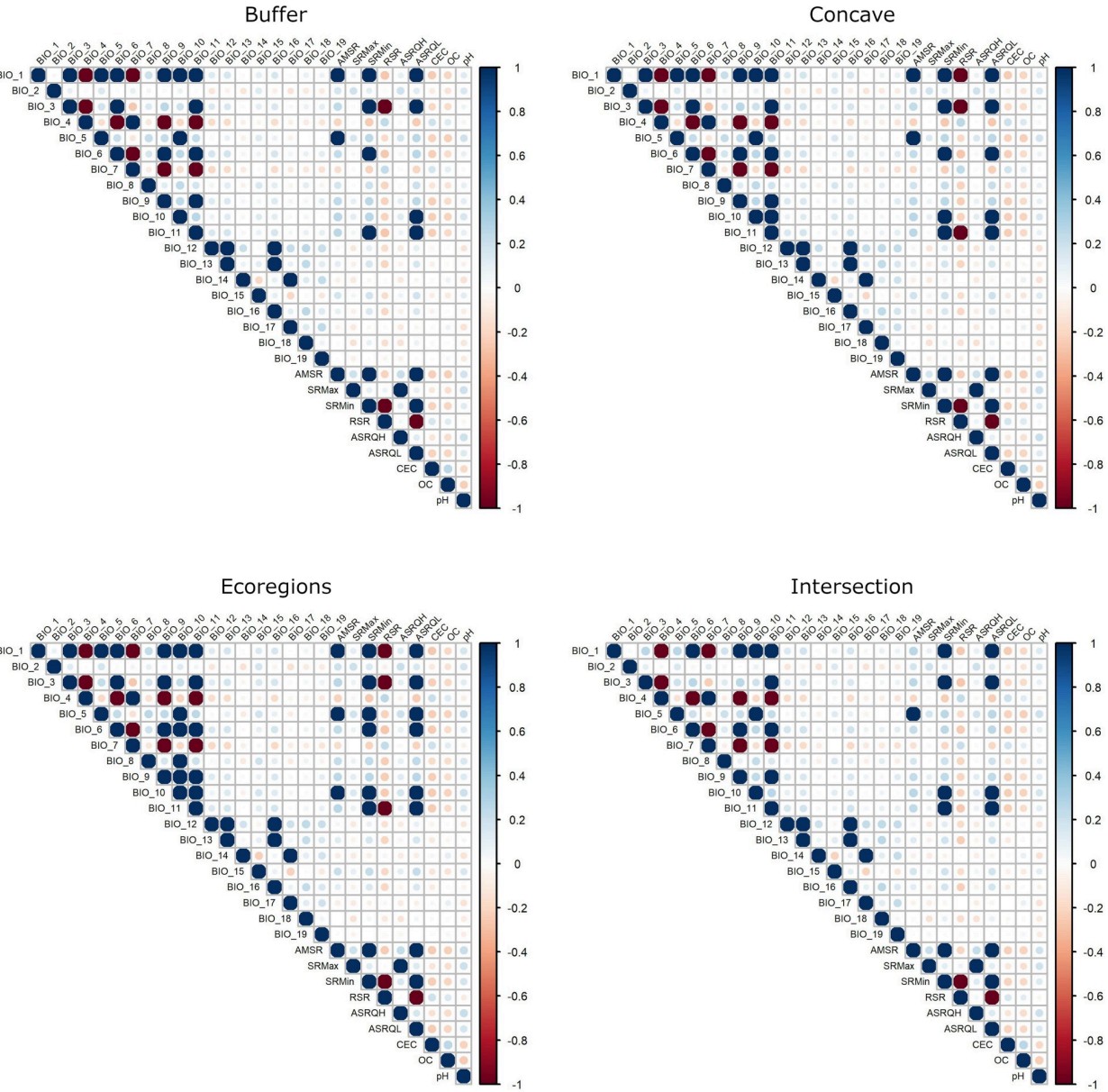

**Fig 3. Results from linear correlation tests for initial variables.** Values of correlation above |0.8| are magnified threefold. Results for variables at 10' resolution are shown. Results for variables at 30' are similar for most variables and can be found in S1 Fig.

## Importance of variables and effects on models

To understand the effect of selected variables on models that could be used to represent species niches and/or potential distributions, we created a final model for each of the calibration areas at the two resolutions of variables, using parameters and sets of variables selected after model calibration. Then, we measured the effects of variables on such models: in Maxent, we used jackknife analysis to measure variable contributions [78], and for GLMs, we used an ANOVA to explore deviance explained by each of the predictors considered and whether deviance values were statistically significant (whether the deviance was larger than expected by chance).

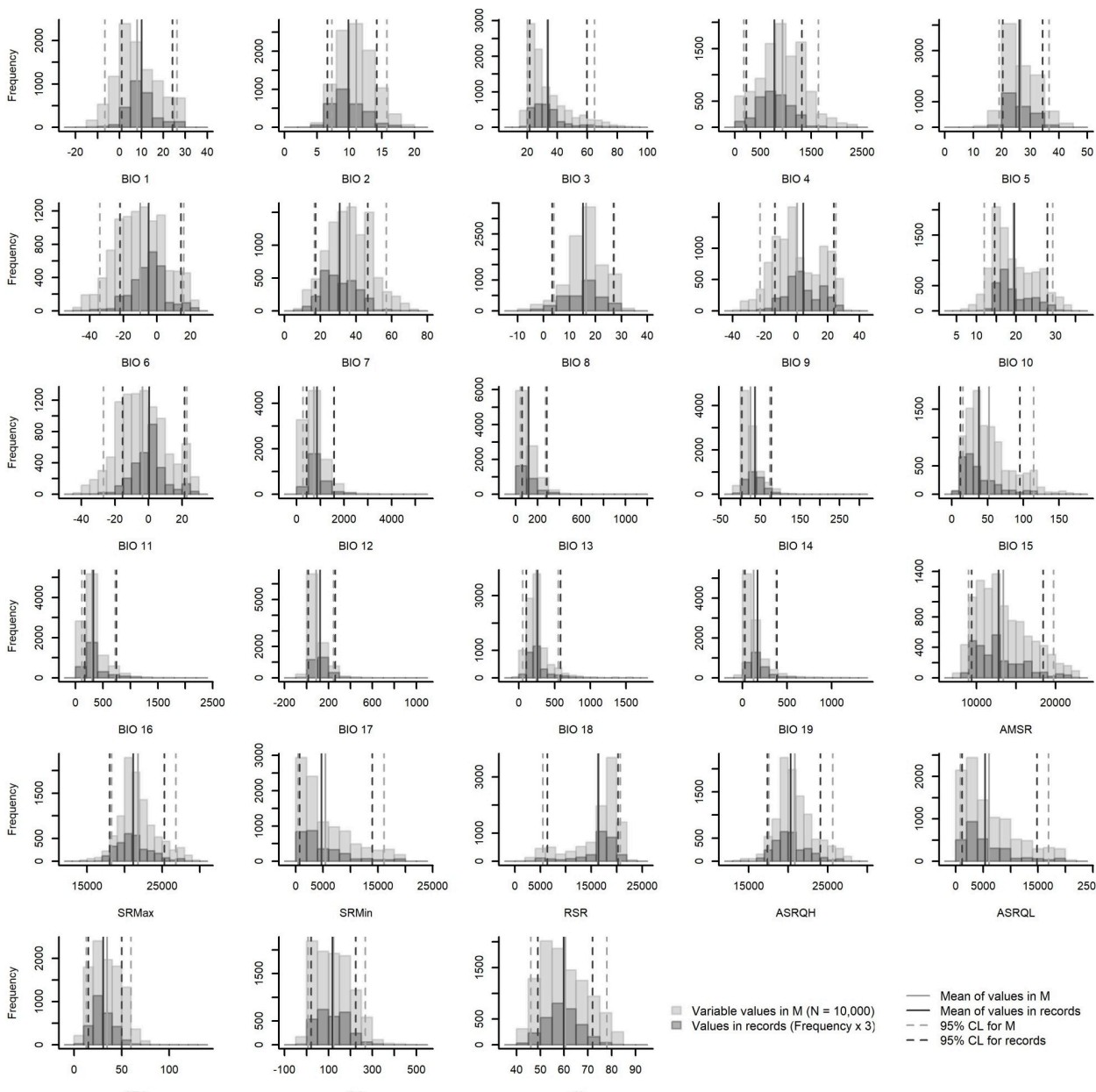

**Fig 4. Histograms of variable values in calibration areas (M) and occurrence records.** Results for variables at 10' resolution and calibration areas resulting from intersection are shown. Results of analyses at 30' resolution and for other calibration areas were similar, although minor differences can be observed in S2–S8 Figs.

We transferred all the models across the area comprising the union of the four calibration areas, and compared those models to assess whether patterns of suitability values differed as a result of using distinct variables, calibration areas, and algorithms. Model transfers in Maxent were done using free extrapolation, no replicates, and a cloglog output format. Model transfers for GLMs were scaled 0–1. As ecological niches exist simultaneously in both geographic and environmental spaces [79], we created 3-dimensional visualizations of resulting predictions in a space defined in terms of some of the environmental variables with larger effects on our

models. Explorations in environmental space were used to detect how variation in suitability was associated with variable values.

## Results

### Results from the selection process

Graphical explorations of environmental conditions across calibration areas and occurrence records varied somewhat among the distinct options of areas for calibration (Figs 4 and S2–S8). Despite such variations, these explorations allowed us to identify variables that appeared better for detecting suitable and unsuitable conditions based on distributions of values and confidence limits. Variable correlations also varied slightly among the distinct options of calibration areas tested, although we consistently found more highly correlated variables in calibration areas derived from ecoregions (Figs 3 and S1). After considering the exploration of environmental conditions, correlation values, and biological importance, we retained 11 variables at 10' resolution and 10 variables at 30' resolution. The variables mean diurnal range (BIO 2), maximum temperature of warmest month (BIO 5), minimum temperature of coldest month (BIO 6), annual precipitation (BIO 12), precipitation of driest month (BIO 14), precipitation seasonality (BIO 15), range of solar radiation (RSR), and average solar radiation of the quarter with highest values (ASRQH), were in common between these sets; cation exchange capacity (CEC), organic carbon (OC), and pH were kept for the set at 10', whereas labile inorganic phosphorus (LIP) and organic phosphorus (OP) were kept at 30".

All model calibration exercises found at least one parameter setting that produced a model that met all criteria for selection (i.e., models with partial ROC values ≤0.05, omission rates ≤0.05, and delta AICc values ≤2; S4 and S5 Tables). Variables selected contrasted markedly among treatments that considered distinct calibration areas, spatial resolutions, and modeling algorithms (Fig 5). None of the final sets of variables selected during model calibration used all of the variables initially selected. In general, fewer variables were selected for models created with Maxent at 10' resolution (2–4) than for the other algorithm/resolution combinations (6–7). Although the subsets of variables considered were not totally comparable between the tests at distinct resolutions, at least one variable representing soil conditions was consistently selected across all exercises using distinct calibration areas, using at least one of the modeling algorithms. Soil and solar radiation variables were more consistently selected at 10' resolution, especially when using Maxent, whereas at 30' resolution, bioclimatic and solar radiation variables were more consistently selected. Bioclimatic and solar radiation variables that represent extreme conditions or means of extreme periods appeared to be selected more consistently regardless of the differences in spatial resolution or algorithm (Fig 5).

### Effects of variables on models

Bioclimatic and solar radiation variables had consistently larger effects than soil variables on Maxent models (S9 and S10 Figs), with the exception of CEC, which was the most important variable for the only model that selected this predictor (i.e., with variables at 10' using calibration areas that intersected the other three options; S9 Fig). The most important predictor for Maxent models varied among BIO variables and CEC at 10' resolution, whereas at 30', BIO 6 was consistently selected as more important based on the contribution, permutation importance, and jackknife results. For GLMs, bioclimatic variables, a few quadratic versions, and products of such variables, as well as CEC, contributed most to the deviance in models at 10' resolution (S6–S9 Tables). Solar radiation variables were not particularly relevant to explain deviance in these models. At 30' resolution, deviance in models was mostly explained by

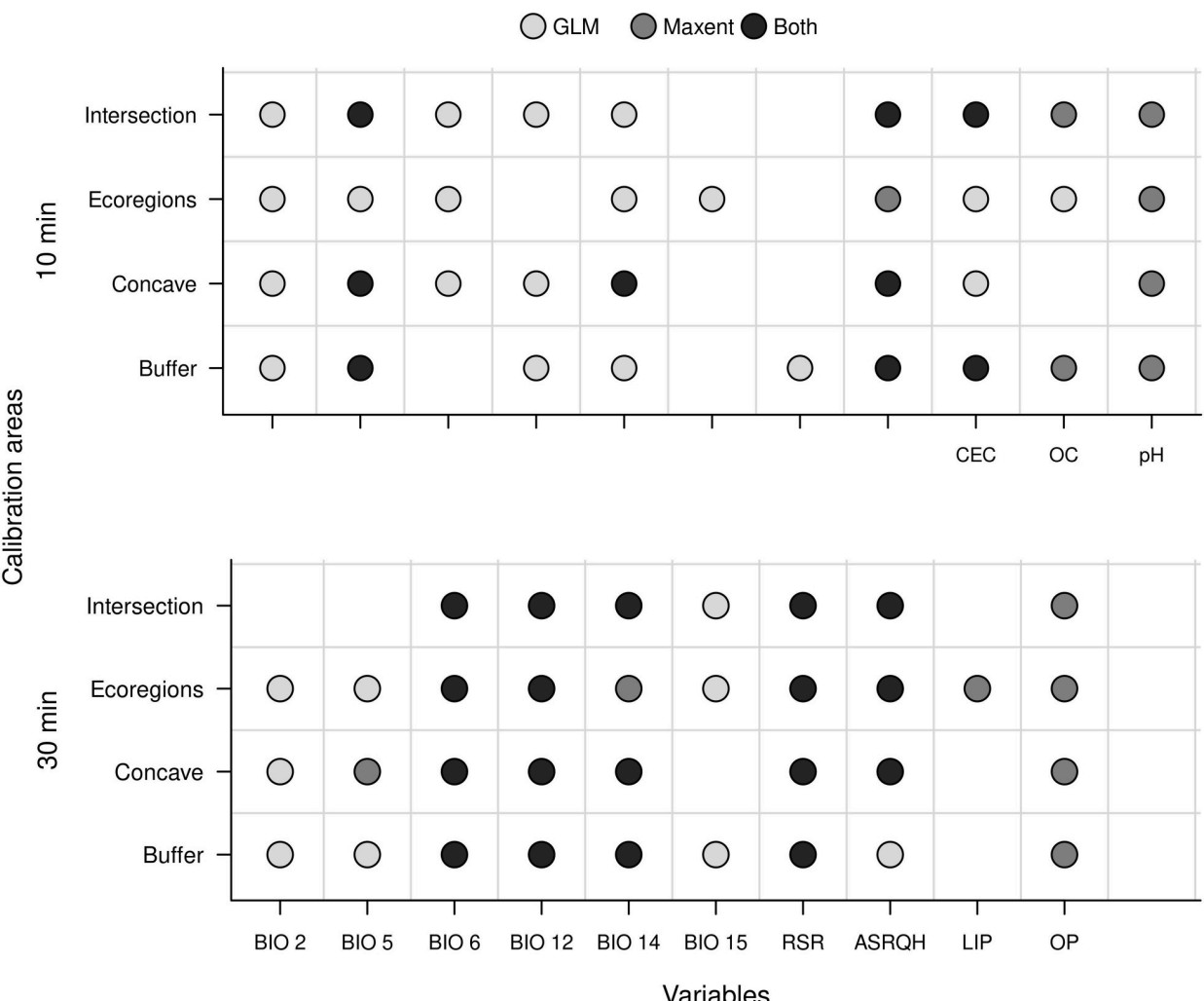

**Fig 5. Summary of variables retained after the multi-step approach for selection.** Results depending on spatial resolution of predictors, model calibration areas, and the algorithm used are shown. BIO2 = mean diurnal range of temperature; BIO5 = maximum temperature of warmest month; BIO6 = minimum temperature of coldest month; BIO12 = annual precipitation; BIO14 = precipitation of driest month; BIO15 = precipitation seasonality; RSR = range of solar radiation; ASRQH = average solar radiation of the quarter with highest values; CEC = cation exchange capacity; OC = organic carbon; LIP = labile inorganic phosphorus; OP = organic phosphorus.

bioclimatic and solar radiation variables, whereas soil variables did not explain large portions of the deviance (S10–S13 Tables).

## Model projections

Geographic transfers of Maxent models at 10' resolution showed higher variability across distinct calibration areas than GLM projections (Figs 6 and S11). Variation was greatest in northern and eastern Asia, central North America, eastern Australia, and northern and southern Africa. At 30' resolution, geographic transfers showed lower variability for both GLM and Maxent models.

Projections of suitability in environmental space showed higher variability in Maxent projections than in GLMs, considering distinct calibration areas at 10' resolution (Figs 7 and S12–S26). That is, suitability values varied highly across the regions of the environment detected as suitable (above the 5% omission threshold). In most Maxent projections of suitability in

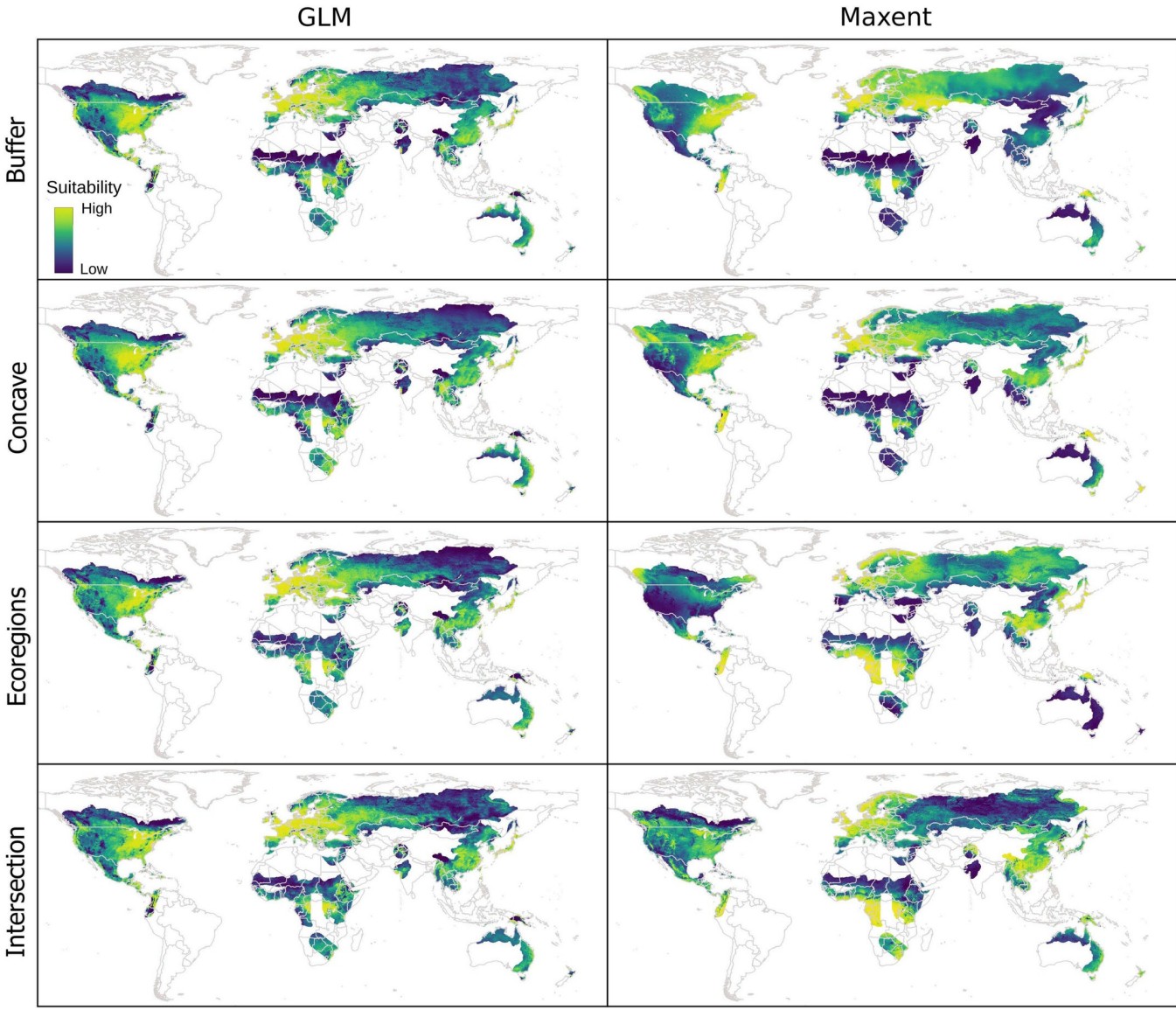

**Fig 6. Geographic projections of suitability values deriving from final models created with the variables selected.** Results for variables at 10' resolution are shown. Results at 30' resolution are presented in S11 Fig.

environmental space, and for various environmental variables, extreme environments were predicted to have high suitability (i.e., we observed truncated responses [2] in our models). GLM projections were more stable in both aspects; in these projections, and considering most variables, regions of high suitability tended to be surrounded by regions with decreasingly lower suitability (i.e., extreme environments were only rarely detected as the most suitable ones). At 30' resolution, projections of environmental space looked similar across distinct calibration areas and modeling algorithms. Perhaps the main difference is that Maxent constrained suitable environments a little more than did the GLMs.

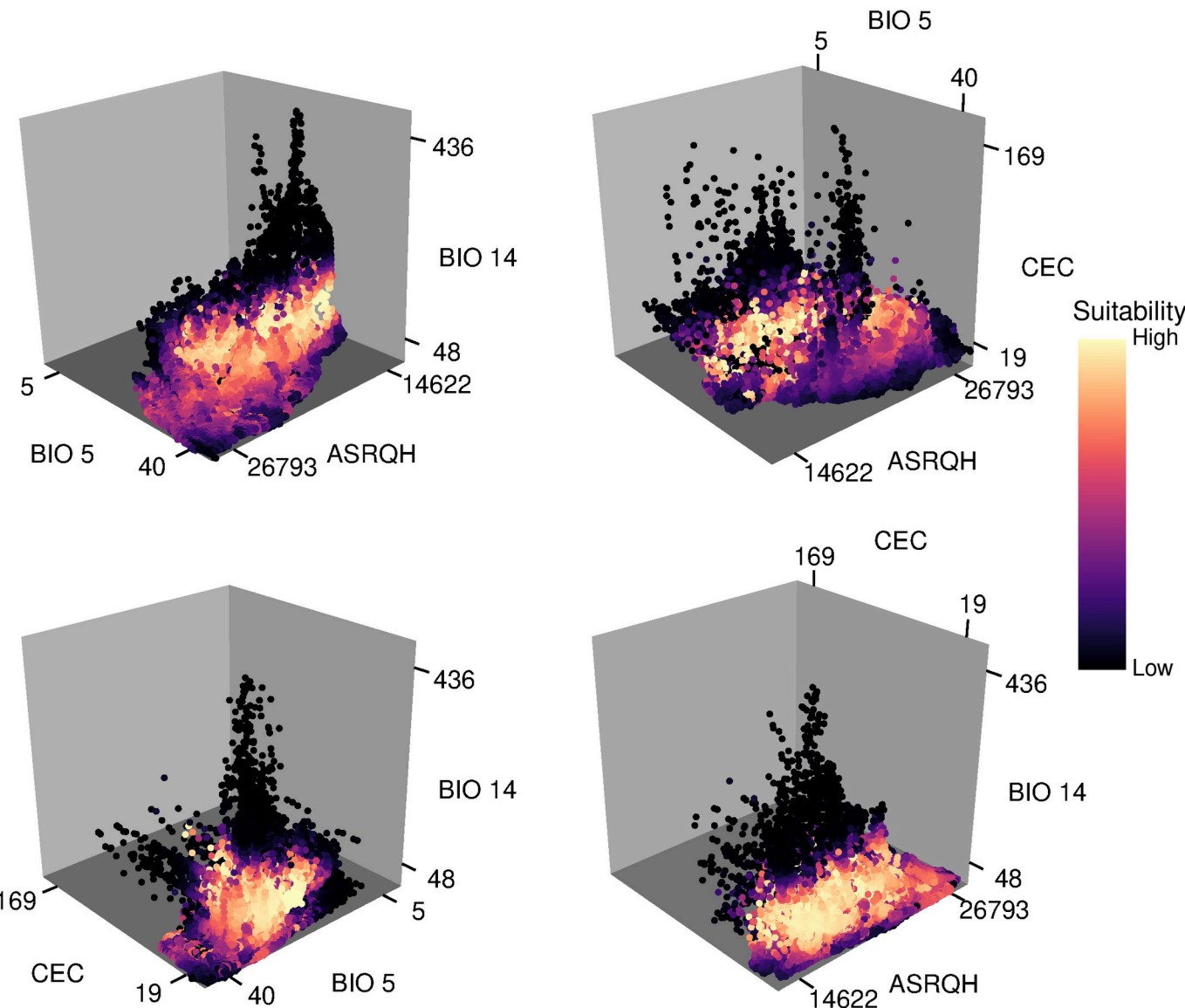

**Fig 7. Projections of suitability values in a three-dimensional environmental space.** Values of suitability derive from final models created with selected variables and parameters. GLM results for variables at 10' resolution and calibration areas resulting from intersection are shown. Results at 30' resolution and for other calibration areas are in S12–S26 Figs.

## Discussion

Determining sets of appropriate environmental variables with which to model ecological niches or potential distributions of species remains a major challenge in distributional ecology. Our results showed that multiple factors generated distinct outcomes regarding which variables are better for model development. Even after an initial selection of relevant variables, when we let statistical methods select the best subset of variables, choice of spatial resolution of layers, area for model calibration, and modeling algorithm, all affected the final subset of predictors selected. However, despite the fact that distinct groups of variables were selected when these factors were changed, some predictors were more consistently selected than others,

which hints at the relevance of such variables when understanding ecological limitations for the species.

Of particular interest is the recurring selection of variables representing maximum temperature (BIO 5) and minimum temperature (BIO 6), which concurs with our initial hypothesis and, in most cases, helped to identify the limits of what is suitable for the species in regions with high and low temperatures. The consistent selection of these two variables could be attributed to the importance of the temperature in the natural history of the species, especially in that low temperatures are responsible for triggering production of turions [25]. If temperatures in an area are consistently low, the species will only produce these dormant fronds, and populations will stop growing (i.e., the species will be outside of its thermal niche, at least for reproduction).

Obtaining different sets of key variables in modeling exercises based on distinct calibration areas is concerning. One of the main aspects to be defined when creating such models is the area over which models will be calibrated [19,59]. This dependency has implications from both statistical and ecological perspectives. From a modeling point of view, the environmental values of the points selected as background or pseudoabsences affect how models are fitted to the occurrence records, sometimes resulting in overfitted models, which complicates model transfers [80,81]. In regard to the ecological relevance of these areas, because models are fitted within these regions, the associations to be found are only relevant if a species has had access to those environmental conditions [18]. As distinct calibration areas affected the set of variables selected and the effects of such variables in the models, correct definition of these areas becomes an even more important challenge. New methods to define calibration areas are now available that account for ecological, historical, and dispersal factors, which may result in more properly calibrated models and more consistent sets of variables [59]. However, this challenge persists in cases in which limited information exists about a species, or the distribution of species is close to global, as in this example.

The other two factors explored (modeling algorithm and spatial resolution of variables) also affected the set of environmental variables selected for niche models. As in previous explorations [22,82,83], the effects of these two factors were seen clearly in the transfers of models, both geographic and environmental, and thus cannot be neglected. Spatial resolution of layers has been noticed as a factor that can influence the sets of variables selected [21]. Depending on the spatial resolution of layers, the number of environmental combinations found in an area can change, with more numerous combinations at finer spatial resolutions. One of the complications deriving from these differences is that the ways in which variables are correlated can change at distinct resolutions due to changes in sample size [84], which can modify the initial selection of variables that is made, not necessarily related to the biological importance of such variables. Our inclusion of distinct modeling algorithms showed that combining these factors certainly increases the complexity of the process of selecting variables. The relevance of distinct variables has been shown to change depending on the algorithm used [22]. Although it is not clear whether the set of variables should be changed depending on the algorithm (if the variables have been preselected in some way), the fact that distinct algorithms work differently and that distinct predictors have distinct effects should not be overlooked [85].

Exploring environmental conditions within calibration areas and in occurrence records beforehand helped to identify variables for which truncated responses could be found. Although this point may not be related to the biological role of this environmental dimension, it is crucial in being able to transfer a model to other conditions with less ambiguity [69,86]. Maximum temperature of warmest month (BIO 5) and minimum temperature of coldest month (BIO 6) are examples of variables that contributed importantly to models, and, as expected, values of suitability were higher at intermediate values, with decreasing suitability

towards extreme environmental values. Performing these explorations can help to select predictors appropriately when the goal is to understand why species are distributed the way they are. However, other variables should not be discarded only based on these graphical explorations, because they may be important environmental constraints despite the truncation. For instance, cation exchange capacity (CEC), a soil variable, showed truncation towards lower values in our examples (Fig 7), but still was selected across various of the experiments, and its contribution to models was not negligible. CEC is a soil variable that provides information about nutrient availability; hence, these results underline the importance of making decisions based on ideas that combine ecological and statistical considerations.

In spite of the variability in the results, we found that variables related to temperature extremes were critical in characterizing the greater duckweed ecological niche, which concurs with findings from experimental work done with this species [25,87]. In fact, temperature may be the main factor shaping the distribution of this species, especially considering its distributional limits at high latitudes. Models created using precipitation variables (particularly those using precipitation of driest quarter; BIO 14) correctly discarded suitability in xeric regions, showing the importance of considering a factor that represents water availability [88]. Solar radiation of quarters periods with higher values (ASRQH) and range of solar radiation (RSR) were also potentially helpful in limiting the distribution of the species towards higher latitudes, as solar radiation informs about a crucial resource for photosynthesis, and experimental work has confirmed the importance of this factor [31,38]. Factors related to soil variables that served as proxies for nutrient availability and water conditions also showed high importance in some of the results. Although nutrients are critical for the development of this species, the fact that soil variables are only indirect proxies for such information [89] and the complications of representing this type of information at the scale of our analyses may explain why these variables were not selected as consistently as others.

In sum, we showed that selecting relevant variables to characterize ecological niches and potential distributions becomes even more complicated when multiple factors related to data processing and model development are considered. However, if a series of criteria and approaches is applied in concert, certain variables are selected more consistently than others. Such variables may in effect be the ones that shape and constrain the species' distribution from a macroecological point of view. Variables representing extreme temperatures, dry periods, seasonality of solar radiation, summer solar radiation, and some soil proxies of nutrients in water were among the factors that contributed the most to shaping the distribution of *S. polyrriza*.

## Supporting information

**S1 Fig. Results from linear correlation tests for initial variables.** Values of correlation above |0.8| are magnified threefold. Results for variables at 30' resolution are shown.
(TIF)

**S2 Fig. Histograms of environmental variable values in calibration areas and occurrence records.** Results for variables at 10' resolution and buffer calibration areas are shown.
(TIF)

**S3 Fig. Histograms of environmental variable values in calibration areas and occurrence records.** Results for variables at 10' resolution and concave calibration areas are shown.
(TIF)

**S4 Fig. Histograms of environmental variable values in calibration areas and occurrence records.** Results for variables at 10' resolution and calibration areas resulting from ecoregions

are shown.
(TIF)

**S5 Fig. Histograms of environmental variable values in calibration areas and occurrence records.** Results for variables at 30' resolution and buffer calibration areas are shown.
(TIF)

**S6 Fig. Histograms of environmental variable values in calibration areas and occurrence records.** Results for variables at 30' resolution and concave calibration areas are shown.
(TIF)

**S7 Fig. Histograms of environmental variable values in calibration areas and occurrence records.** Results for variables at 30' resolution and calibration areas resulting from ecoregions are shown.
(TIF)

**S8 Fig. Histograms of environmental variable values in calibration areas and occurrence records.** Results for variables at 30' resolution and calibration areas resulting from intersection are shown.
(TIF)

**S9 Fig. Predictor contribution to Maxent models created with variables and parameter settings selected after model calibration.** Results for variables at 10' resolutions are shown.
(TIF)

**S10 Fig. Predictor contribution to Maxent models created with variables and parameter settings selected after model calibration.** Results for variables at 30' resolutions are shown.
(TIF)

**S11 Fig. Geographic projections of suitability values deriving from final models created with the selected variables.** Results for variables at 30' resolution are shown.
(TIF)

**S12 Fig. Projections of suitability values in a three-dimensional environmental space.** Values of suitability derive from final models created with selected variables and parameters. GLM results for variables at 10' resolution and calibration areas resulting from buffers are shown.
(TIF)

**S13 Fig. Projections of suitability values in a three-dimensional environmental space.** Values of suitability derive from final models created with selected variables and parameters. GLM results for variables at 10' resolution and calibration areas resulting from concave hulls are shown.
(TIF)

**S14 Fig. Projections of suitability values in a three-dimensional environmental space.** Values of suitability derive from final models created with selected variables and parameters. GLM results for variables at 10' resolution and calibration areas resulting from ecoregions are shown.
(TIF)

**S15 Fig. Projections of suitability values in a three-dimensional environmental space.** Values of suitability derive from final models created with selected variables and parameters. GLM results for variables at 30' resolution and calibration areas resulting from buffers are

shown.
(TIF)

**S16 Fig. Projections of suitability values in a three-dimensional environmental space.** Values of suitability derive from final models created with selected variables and parameters. GLM results for variables at 30' resolution and calibration areas resulting from concave hulls are shown.
(TIF)

**S17 Fig. Projections of suitability values in a three-dimensional environmental space.** Values of suitability derive from final models created with selected variables and parameters. GLM results for variables at 30' resolution and calibration areas resulting from ecoregions are shown.
(TIF)

**S18 Fig. Projections of suitability values in a three-dimensional environmental space.** Values of suitability derive from final models created with selected variables and parameters. GLM results for variables at 30' resolution and calibration areas resulting from intersection are shown.
(TIF)

**S19 Fig. Projections of suitability values in a three-dimensional environmental space.** Values of suitability derive from final models created with selected variables and parameters. Maxent results for variables at 10' resolution and calibration areas resulting from buffers are shown.
(TIF)

**S20 Fig. Projections of suitability values in a three-dimensional environmental space.** Values of suitability derive from final models created with selected variables and parameters. Maxent results for variables at 10' resolution and calibration areas resulting from concave hulls are shown.
(TIF)

**S21 Fig. Projections of suitability values in a three-dimensional environmental space.** Values of suitability derive from final models created with selected variables and parameters. Maxent results for variables at 10' resolution and calibration areas resulting from ecoregions are shown.
(TIF)

**S22 Fig. Projections of suitability values in a three-dimensional environmental space.** Values of suitability derive from final models created with selected variables and parameters. Maxent results for variables at 10' resolution and calibration areas resulting from intersection are shown.
(TIF)

**S23 Fig. Projections of suitability values in a three-dimensional environmental space.** Values of suitability derive from final models created with selected variables and parameters. Maxent results for variables at 30' resolution and calibration areas resulting from buffers are shown.
(TIF)

**S24 Fig. Projections of suitability values in a three-dimensional environmental space.** Values of suitability derive from final models created with selected variables and parameters.

Maxent results for variables at 30' resolution and calibration areas resulting from concave hulls are shown.
(TIF)

**S25 Fig. Projections of suitability values in a three-dimensional environmental space.** Values of suitability derive from final models created with selected variables and parameters. Maxent results for variables at 30' resolution and calibration areas resulting from ecoregions are shown.
(TIF)

**S26 Fig. Projections of suitability values in a three-dimensional environmental space.** Values of suitability derive from final models created with selected variables and parameters. Maxent results for variables at 30' resolution and calibration areas resulting from intersection are shown.
(TIF)

**S1 Table. Spatial autocorrelation results for all environmental variables derived from spatial patterns of occurrence data after using distinct distances for spatial thinning.** Results presented here are for variables at 10' resolution. Spatial autocorrelation was measured using the statistic Moran's I.
(DOCX)

**S2 Table. Spatial autocorrelation results for all environmental variables derived from spatial patterns of occurrence data after using distinct distances for spatial thinning.** Results presented here are for variables at 30' resolution. Spatial autocorrelation was measured using the statistic Moran's I.
(DOCX)

**S3 Table. Description of ecological importance of variables used for ecological niche modeling exercises with *Spirodela polyrhiza*.**
(DOCX)

**S4 Table. Selected parameter settings and variables after model calibration for analyses with variables at 10' resolution.** AIC/AICc values are not comparable across distinct calibration areas.
(DOCX)

**S5 Table. Selected parameter settings and variables after model calibration for analyses with variables at 30' resolution.** AIC/AICc values are not comparable across distinct calibration areas.
(DOCX)

**S6 Table. Effects of predictors on GLMs produced using variables and parameter settings selected after model calibration.** Results for models created with variables at 10' resolution, using buffer calibration areas are shown. Quadratic = "^2"; Product = ":".
(DOCX)

**S7 Table. Effects of predictors on GLMs produced using variables and parameters settings selected after model calibration.** Results for models created with variables at 10' resolution, using concave calibration areas are shown. Quadratic = "^2"; Product = ":".
(DOCX)

**S8 Table. Effects of predictors on GLMs produced using variables and parameters settings selected after model calibration.** Results for models created with variables at 10' resolution,

using calibration areas from ecoregions are shown. Quadratic = "^2"; Product = ":".
(DOCX)

**S9 Table. Effects of predictors on GLMs produced using variables and parameters settings selected after model calibration.** Results for models created with variables at 10' resolution, using calibration areas from intersection are shown. Quadratic = "^2"; Product = ":".
(DOCX)

**S10 Table. Effects of predictors on GLMs produced using variables and parameters settings selected after model calibration.** Results for models created with variables at 30' resolution, using buffer calibration areas are shown. Quadratic = "^2"; Product = ":".
(DOCX)

**S11 Table. Effects of predictors on GLMs produced using variables and parameters settings selected after model calibration.** Results for models created with variables at 30' resolution, using concave calibration areas are shown. Quadratic = "^2"; Product = ":".
(DOCX)

**S12 Table. Effects of predictors on GLMs produced using variables and parameters settings selected after model calibration.** Results for models created with variables at 30' resolution, using calibration areas from ecoregions are shown. Quadratic = "^2"; Product = ":".
(DOCX)

**S13 Table. Effects of predictors on GLMs produced using variables and parameters settings selected after model calibration.** Results for models created with variables at 30' resolution, using calibration areas from intersection are shown. Quadratic = "^2"; Product = ":".
(DOCX)

## Acknowledgments

MEC thanks his doctoral dissertation committee for the initial suggestion of this set of analyses. We thank the members of the KUENM working group in the University of Kansas Biodiversity Institute, for their thinking and work on these topics over the years.

## Author Contributions

**Conceptualization:** Marlon E. Cobos, A. Townsend Peterson.

**Data curation:** Marlon E. Cobos.

**Formal analysis:** Marlon E. Cobos.

**Funding acquisition:** A. Townsend Peterson.

**Investigation:** Marlon E. Cobos, A. Townsend Peterson.

**Methodology:** Marlon E. Cobos, A. Townsend Peterson.

**Writing – original draft:** Marlon E. Cobos.

**Writing – review & editing:** Marlon E. Cobos, A. Townsend Peterson.

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
