## [Decision Letter · Decision Letter 0]

31 Jan 2023

PONE-D-22-28513Broad-scale factors shaping the ecological niche and geographic distribution of Spirodela polyrhizaPLOS ONE

Dear Dr. Cobos,

Thank you for submitting your manuscript to PLOS ONE. After careful consideration, we feel that it has merit but does not fully meet PLOS ONE’s publication criteria as it currently stands. Therefore, we invite you to submit a revised version of the manuscript that addresses the points raised during the review process. Most of the referees found merit in this manuscript, especially pointing out the accurate framework implemented by the authors to select optimal environmental variables for model calibration. That said, some points need to be addressed before proceeding further with this manuscript. Firstly, hypotheses and objectives must be clearly specified as to put the study significance in appropriate evidence. Also, important details about methodological choices miss from the text (e.g. the rationale behind pseudo-absences placement and number, information on the modelling algorithms, etc), as well as ecological justification for the sizes of the calibration area or the distance considered to reduce sampling bias. Regarding this latter point, the authors should also make sure that the spatial autocorrelation in models' residuals was actually absent or not significant.

We look forward to receiving your revised manuscript.

Kind regards,

Mirko Di Febbraro

Academic Editor

PLOS ONE

Journal Requirements:

2. "Please upload a new copy of Figure 2 and 4 as the detail is not clear. Please follow the link for more information: " ext-link-type="uri" xlink:type="simple">https://blogs.plos.org/plos/2019/06/looking-good-tips-for-creating-your-plos-figures-graphics/"" " ext-link-type="uri" xlink:type="simple">https://blogs.plos.org/plos/2019/06/looking-good-tips-for-creating-your-plos-figures-graphics/"

Reviewers' comments:

Reviewer's Responses to Questions

**Comments to the Author**

1. Is the manuscript technically sound, and do the data support the conclusions?

Reviewer #1: Yes

Reviewer #2: No

Reviewer #3: Yes

Reviewer #4: Partly

2. Has the statistical analysis been performed appropriately and rigorously? 

Reviewer #1: Yes

Reviewer #2: No

Reviewer #3: Yes

Reviewer #4: Yes

3. Have the authors made all data underlying the findings in their manuscript fully available?

Reviewer #1: Yes

Reviewer #2: Yes

Reviewer #3: Yes

Reviewer #4: Yes

4. Is the manuscript presented in an intelligible fashion and written in standard English?

Reviewer #1: Yes

Reviewer #2: No

Reviewer #3: Yes

Reviewer #4: Yes

5. Review Comments to the Author

Reviewer #1: The interesting paper entitled “Broad-scale factors shaping the ecological niche and geographic distribution of Spirodela polyrhiza” showed a multi-step approach to select relevant variables for modelling the ecological niche. A widely distributed aquatic plant species was used as a case study, but this approach can be widely replicated for different species, geographic regions and scales. The analysis led to interesting and reliable results on plant-specific ecology. Further thoughts on how underestimated factors, such as the calibration area and spatial resolution used, greatly influence the response of each model are also of interest.

Overall, I consider this manuscript suitable for publication in Plos One.

I have only a few concerns:

I suggest reporting the authority of the species when first mentioned.

L 136-137: the authors mention that to reduce autocorrelation bias, they thinned the records using a minimum point-to-point distance of ~30'. It would be useful to test and compare this autocorrelation to show and then discuss whether it was indeed reduced.

L 170: please cite the source from which the ecoregions were retrieved.

Attachments: for ease of reading, I suggest changing the format of the supporting figures (e.g. jpeg) or directly merging all this information (including descriptions) into one pdf file.

Reviewer #2: The main aim of the MS is to identify the abiotic factors that shape the distribution of Spirodela polyrhiza. This aim could be crucial to understand and predict the actual and future distribution of the target species, but in my opinion the methods used are completely wrong:

First of all raster resolution of explanatory variables at 10’ or 30’ cannot be acceptable, especially to investigate the distribution of a species that lives in the ponds.

Secondly, the authors try to disentangle some methodological issues in SDM/ENM (e.g. extend of the calibration areas, algorithms) that cannot be addressed using very coarse variables and considering only one species.

Specific methodological comment:

- Records that were outside of, but closer than ~5’ to the edge of environmental layers were moved to the nearest pixel with information – what? are you expected to find similar environmental condition within 100 km2

- minimum point-to-point distance of ~30’ – how many different environmental condition occurs in ~3600 km2 - not acceptable

- the broad distribution of this species makes it difficult for that method to be applied, why? Maybe is due to the coarse resolution of your approach?

- Modeling algorithms: how did you generate the pseudoabsence? how did you calibrate the models,? have you performed some how the cross validation?

- “We transferred all the models across the area comprising the union of the four calibration areas and compared those models to assess whether patterns of suitability values differed as a result of using distinct variables, calibration areas, and algorithms.” WHY? Is it a methodological or ecological MS? Not clear.

- Have you considered the multicollinearity among explanatory variables?

Please consider to read many paper that suggest how perform the SDM/ENM, how create pseudo-absence, how to create ensemble models by combining different algorithms, how to perform cross validation and so on. Please not use a single species to address methodological question, consider to use virtual species for that. Hereunder some useful paper:

Barbet‐Massin, M., Jiguet, F., Albert, C. H., Thuiller, W. (2012). Selecting pseudo‐absences for species distribution models: how, where and how many?. Methods in ecology and evolution, 3(2), 327-338.

Bucklin, D. N., Basille, M., Benscoter, A. M., Brandt, L. A., Mazzotti, F. J., Romanach, S. S., ... Watling, J. I. (2015). Comparing species distribution models constructed with different subsets of environmental predictors. Diversity and distributions, 21(1), 23-35.

Connor, T., Hull, V., Viña, A., Shortridge, A., Tang, Y., Zhang, J., ... Liu, J. (2018). Effects of grain size and niche breadth on species distribution modeling. Ecography, 41(8), 1270-1282.

Muscarella, R., Galante, P. J., Soley‐Guardia, M., Boria, R. A., Kass, J. M., Uriarte, M., Anderson, R. P. (2014). ENM eval: An R package for conducting spatially independent evaluations and estimating optimal model complexity for Maxent ecological niche models. Methods in ecology and evolution, 5(11), 1198-1205.

VanDerWal, J., Shoo, L. P., Graham, C., Williams, S. E. (2009). Selecting pseudo-absence data for presence-only distribution modeling: how far should you stray from what you know?. Ecological modelling, 220(4), 589-594.

Reviewer #3: The paper entitled "Broad-scale factors shaping the ecological niche and geographic distribution of Spirodela polyrhiza", investigates the effect of variable selection procedures for modeling the ecological niche of the aquatic Spirodela polyrriza, taking into account variability arising from using distinct algorithms, calibration areas, and spatial resolutions of variables. The authors show that the final set of variables selected based on statistical inference varied considerably depending on the combination of algorithm, calibration area, and spatial resolution used.

The article is clearly written, polished, well-edited, and scientifically sound. It is characterized by good originality. I recommend publishing with minor changes. I have a few comments.

Abstract

The abstract accurately describes the main objective of the study. It explains how the study was done, including the model organism used, without exceeding methodological details. The most important results are summarized, but their significance has not been sufficiently emphasized, specifically for the variation of the final set of variables selected based on the combination of algorithm, calibration area, and spatial resolution.

Introduction

The authors provide a careful overview of the challenge of selecting appropriate environmental variables when characterizing species' ecological niches, and what still ought to be done. The objectives of the study are clearly specified, but hypothesis are missing. The lack of clear hypothesis could prevent to really understand the significance and the importance of the study.

Methods

The Materials and Methods section provides enough detail to allow suitably skilled investigators to replicate the main steps of the study. However, the lack of specific information (e.g., method of generating pseudo-absences, method details of GLMs' model calibration, R packages adopted to evaluate GLMs' performance of candidate models) does not allow a full understanding of the code provided. Specific information should be included in detail, citing articles you followed for the choices/methods applied.

In addition, I suggest the authors explain why they adopted that specific ratio of the quantity of presence data to the number of background points/pseudo-absence data to fit models, specifying a reference that supports the choice. This is always a very sore point because according to some authors an inadequate number of background points or pseudo-absences (in this study 20,000 points for 964 occurrences; lines 138 and 190-191) could reduce the accuracy of the models. Model accuracy is generally affected both by this ratio, but also by the method used to generate pseudo-absences (see for example

Barbet-Massin et al., 2012 -

https://besjournals.onlinelibrary.wiley.com/doi/full/10.1111/j.2041-210X.2011.00172.x).

More specific comments here:

Line 136 - 138: I suggest the authors provide details on the reference that supports the choice of using the minimum point-to-point distance of ~30’ for the spatial thinning of species occurrence records.

Line 160 - 162: "We performed raster aggregation procedures (average of values) on CEC, OC, and pH to match the resolution of BIO variables, and on BIO and SR variables to match the resolution of variables at 30’. [..]" I do not really understand this sentence. Have you used a BIO variable as a snap raster to ensure all cells were properly aligned, and all rasters have the same cell resolutions? If this is the case, please specify the method applied (e.g., nearest neighbour method etc.), if not rephrase the sentence.

Line 162 - 164: "Although the set of variables representing soil conditions used at 10’ differs from the one at 30’, variable selection analyses will help to identify whether the variables selected differ between the two resolutions. [..]"

Does any other author support this??

Line 171 - 176: "Although a new simulation-based approach has been recently suggested as a reliable tool to estimate calibration areas [59], the broad distribution of this species makes it difficult for that method to be applied. Our chosen calibration areas are therefore reasonable options to calibrate models, considering that such areas should reflect what regions could have been accessible to the species and present relevant environments for comparisons [..]"

Does any other author support this?? Do you have a specific reference to cite?

Line 216-217: "The latter consideration assumes that using variables for which the entire spectrum of responses can be characterized makes for better models. [..]" Please cite and provide a reference for this sentence.

Line 218-219: "Biological relevance of variables was determined based on details about the species' natural history, phenology, and physiology in the literature, and our own experience with populations in the field and controlled environments [..]" No bibliographic reference was cited to justify the selection of biologically relevance variables. Please, integrate a citation

Line 260: “see below”. Where? Please, specify.

Results

The results relate to the research question, and the language adopted to express results is clear and concise. The tables and figures are appropriate, but very fragmented in many appendix documents. To explore the results in detail, the reader should open as many as 11 documents (.docx format) and have vector graphic software to open 26 images attached in the .eps format. I strong suggest aggregating appendixes and exporting images in a simple file format (e.g., jpg or .tif).

Discussion

The writing is very good and well-polished. I have no suggestion.

Reviewer #4: General comments

The authors of “Broad-scale factors shaping the ecological niche and geographic distribution of Spirodela polyrhiza” focused their research to define an innovative methodological approach in order to select the most appropriate set of environmental variables in ecological niche modeling. This aim is the basis for produce efficient ecological niche modeling, and even today it still not fully resolved. Currently the traditional methods to establish the set of environmental variables consists in different approach for example excluding the variables with very high multicollinearity problems, selecting the variable through expert based procedure supported by empirical evidences, and also letting algorithms that they eliminate the variables in order to optimize the fitting of ecological niche model produced. Furthermore as well described by authors, the final set of environmental variables in ecological niche modeling depends on many different factors. Firstly in the ecological niche modelling, these models can be produced to respond two different aims to analyze the ecology of species target and its environmental limits or to predict the geographic distribution of species target, and consequently these two different aims require to define dissimilar environmental variables. Moreover the set of environmental variables may varies depending on spatial resolution of environmental data used, on areas for model calibration, and also on algorithm used. These three last aspects will require further examines given that there are still few researches that they directly investigated these questions. In this context, the authors developed ecological niche models for Spirodela polyrhiza, a cosmopolitan free-floating aquatic plants on different calibration areas, using environmental variables at different spatial resolutions, using two niche model algorithms and also applying a multi-step approach to define the environmental. The occurrence data were downloaded from GBIF and Botanical Information and Ecology Network at global scale, successively these data were filtered maintaining 964 occurrences. The authors used environmental variables with different spatial resolution, in particular bioclimatic and solar radiation at 10’ of resolution acquired from WorldClim v2.1, soil variables as cation exchange capacity, organic carbon and ph from World Soil Information database at fine resolution of 250 m and coarser soil variables as total phosphorous, labile inorganic phosphorous and organic phosphorous at 30’ resolution. Furthermore, the authors considered four different areas for model calibration: first area was defined as buffers of 5° around S. polyrhiza occurrences, the second area consist in concave-hull polygons with a buffer of 5° around S. polyrhiza occurrences, the third area was the ecoregions occupied by the species buffered by 1° and finally the fourth area was the intersection of the previous three areas. Concerning the ecological niche model algorithms the authors calculated the generalized linear models (GLM with different weight, 1 for S. polyrhiza occurrences and 10,000 for pseudo-absence) and Maxent using 20,000 pseudo-absence and background data respectively. Finally, the environmental variables were selected using a multi-step approach that well summarize a large number of qualitative and quantitative approaches individually applied on previous studies. First step (that is only described in Figure 2) consist in inspection and/or treatment of variables, after a measure of linear correlation among variables, followed by exploration of variable values inside and outside the S. polyrhiza occurrence areas. At the end of the third step, the authors proposed a first selection of variables supported by ecological and historic information of species target. After, the variables were assembled between them in all combinations from two to total number of variables. Finally all these dataset were used in ecological niche models with different calibration areas and algorithms (GLM, Maxent) with a total of 10,180 and 5065 GLM models were tested at spatial resolution of 10’ and 30’ respectively and also for Maxent algorithms were produced 61,080 and 30,390 niche models at 10’ and 30’ respectively produced. The performance of each models was calculated using the following metrics: partial ROC, omission rate and Akaike information criterion for GLMs, and the AICc for Maxent. Ultimately, the authors defined the effects of environmental variables analyzing the best model for each algorithms (GLM, Maxent) and calibration area and in the two spatial resolutions (10’, 30’) through the use of jackknife analysis for Maxent and ANOVA for GLM.

The results of this research demonstrated the potentiality of this approach to define the best set of environmental variables. Firstly, the graphical explorations and the linear correlation of environmental conditions across calibration areas and S. polyrhiza occurrences enables to display the variables with higher suitable conditions in order to reduce at 11 variables with 10’ of spatial resolution and 10 variables with 30’ of spatial resolution. Interestingly, the differences of environmental variables selected at different resolution, concerning 10’ resolution the soil and solar radiation variables were more suitable to analyze the S. polyrhiza occurrences whereas at 30’ resolution the bioclimatic and solar radiation variables were more suitable. Finally, the Maxent algorithm seems to work better compared to GLM given that the Maxent projections showed higher variability across for each spatial resolution and the calibration areas.

In general, this research the authors very well examine the problem due to the selection of environmental variables for niche ecological modelling. The manuscript is well structured in particular in introduction, methods and discussion, less the results that it requires a large number of information as the results of model evaluation (partial ROC, omission rate, Akaike information criterion for GLM and AICc for Maxent), and also the results of jackknife analysis for Maxent and ANOVA for GLM to measure/explore deviances for each environmental variables selected. Moreover, the results obtained for S. polyrhiza niche models are in line with large number of previous researches with different species target in which the bioclimatic, solar radiation and soil conditions variables were identified as the most important environmental variables that limit species growth at global scale. Consequently in the manuscript, miss a clear paragraph that describe innovative aspects to use these environmental variables in order to produce S. polyrhiza niche models. I consider that the use of traditionally methods to define the set of environmental variables allowed to achieve same or similar results. Please, provide you a motivation for this my question. Furthermore, several steps in text require more details and adequate motivations, for example miss ecological reason of dimensions and shapes of calibration areas, miss ecological reason of distance (30’) required to reduce the bias of spatial autocorrelation, and also miss the ecological reasons for choosing these environmental variables (bioclimatic, solar radiation, and soil conditions, for more details see Specific comments). Finally, the use of GBIF and BIEN underestimate the real spatial extension of S. polyrhiza. Can this problem effect the ecological niche model produced?

In consideration of above, this paper may be addressed in a major revision.

Specific comments

Line 10: Change the corresponding author email with an institutional email.

Lines 134 – 136: I did not understand this sentence. Are environmental variables not at global scale? Please rewrite this sentence, thank you.

Lines 137 – 138: The S. polyrhiza occurrences were subjected to a drastic reduction from 45,913 to 964 records. This reduction is due to minimum point-to-point distance (30’) set to eliminate the spatial autocorrelation problem. Please, you include in the manuscript an ecological motivation specific to S. polyrhiza to establish this distance.

Paragraph “Environmental variables”: In this paragraph misses a description how these environmental variables were important to S. polyrhiza growth.

Lines 157 – 160: Add a table with the environmental variables used describing their ecological importance specific to S. polyrhiza and the spatial resolution.

Lines 167 – 171: Please add references that they used similar methods to define calibration area, and also indicated the importance of these methods (buffer, concave-hull polygons, ecoregions, and the intersection of previous three areas) as calibration areas for S. polyrhiza.

Paragraph “Modeling algorithms”: Even if the calibration areas include large portion of territory around S. polyrhiza occurrences, these occurrences derived by GBIF and BIEN dataset that could be underestimated the areal of target species. Consequently, background or pseudo-absence points could be false negative. How you considered this question?

Lines 198 – 200 and Figure 2: In the main text miss a description of the first step displayed in Figure 2 “Inspection and/or treatment”. Furthermore, please add the same number for each step reported in main text in figure 2 to make the reading of manuscript easier.

Lines 212 – 216: The references of Fig. 3 and Fig. 4 were inverted in main text. After, these two figures (Fig. 3, Fig. 4) showed preliminary results of S. polyrhiza ecological niche model. In my researches, I prefer described these results in the results paragraph. Please move these two figures and add their description in results paragraph.

Figures 3, 4, 6, 7: Add in the main text or in the caption of these figures the reasons to display in the main text only the results with 10’ in spatial resolution.

Figures 6 and S11: Add a legend of suitability.

Lines 367 – 370: Based on this sentence, I have a question. The S. polyrhiza occurrences used in this research could be not include the total real occurrences, can this problem influenced the results of ecological niche models?

Lines 403 – 408: Add ecological reasons specific and references to S. polyrhiza and/or other species in order to motivate the inclusion of variables with complication in graphical explorations.

Lines 411 – 413: The temperature is certainly the main driver to limit the species growth at global scale, please add references.

Lines 419 – 421: The relationship between the target species and soil variables were more important in the finest ecological niche model. This describe an higher importance of soil condition at this fine scale, please add references to motivate this feature.

6. PLOS authors have the option to publish the peer review history of their article (what does this mean?). If published, this will include your full peer review and any attached files.

Reviewer #1: **Yes: **Mauro Fois

Reviewer #2: No

Reviewer #3: No

Reviewer #4: No

---

## [Author Response · Author response to Decision Letter 0]

10 Apr 2023

Major comment from Editor and Reviewers:

Editor: Regarding this latter point (distance considered to reduce sampling bias), the authors should also make sure that the spatial autocorrelation in models' residuals was actually absent or not significant.

Reviewer 1: L 136-137: the authors mention that to reduce autocorrelation bias, they thinned the records using a minimum point-to-point distance of ~30'. It would be useful to test and compare this autocorrelation to show and then discuss whether it was indeed reduced.

Reviewer 2: … minimum point-to-point distance of ~30’ – how many different environmental condition occurs in ~3600 km2 - not acceptable

Reviewer 3: Line 136 - 138: I suggest the authors provide details on the reference that supports the choice of using the minimum point-to-point distance of ~30’ for the spatial thinning of species occurrence records.

Reviewer 4: … miss ecological reason of distance (30’) required to reduce the bias of spatial autocorrelation. Lines 137 – 138: The S. polyrhiza occurrences were subjected to a drastic reduction from 45,913 to 964 records. This reduction is due to minimum point-to-point distance (30’) set to eliminate the spatial autocorrelation problem. Please, you include in the manuscript an ecological motivation specific to S. polyrhiza to establish this distance.

Response: We understand this concern and are aware of the usual problems related to sampling bias and how spatial thinning can help to reduce this problem. We consider that spatial correlation in model residuals are still going to exist after our thinning process. However, despite not solving the issue of spatial autocorrelation completely, the distance filter that we used helps to exclude duplicated information and reduce problems derived from autocorrelation without losing more data. We have tried distinct filter distances (0, 50, 100, 150, 200, and 250 km) to handle spatial autocorrelation issues in our study (measured using Moran’s I), but none of these options solves the problem entirely (S1-S2 Tables). That is, after using our thinning distance (30’, ~50 km) observed values of Moran’s I are reduced by orders of magnitude for all predictors; these values do not decrease appreciably more with increased thinning distances. In fact, none of the distances tested erases the problem of spatial autocorrelation (i.e., all P-values were 0.01); the loss of data increases with larger distances. We believe that this is happening because of the spatial arrangement of our data, which are almost global in extent but are biased towards the northern parts of the planet. We also consider that the intensive model selection process that we used (aiming to meet statistical significance, omission, and model complexity criteria) helps to reduce the effect of spatial autocorrelation in our results.

We respectfully disagree with the statement that 30’ is a non-acceptable distance for such filtering. Analyses in ecological niche modeling can be done at multiple scales and grain size. As we are explicitly acknowledging that our analyses are coarse, the uncertainty of coordinates in many places in Europe is high, and our conclusions are directed to broad-scale aspects, we consider that this distance is justified. 

We have modified the text in the manuscript to explain our motivation to use this distance (see lines 139-142), and included the table below as part of our supplementary materials.

Reviewer #1: 

I suggest reporting the authority of the species when first mentioned.

Response: We added the authority as suggested (see line 92).

L 170: please cite the source from which the ecoregions were retrieved.

Response: No particular reference is associated with this layer, however, we have added a reference to the database from which we obtained such data (see line 192).

Attachments: for ease of reading, I suggest changing the format of the supporting figures (e.g. jpeg) or directly merging all this information (including descriptions) into one pdf file.

Response: We included the supporting figures and tables as a separate PDF file for easy reading. However, following the journal requirements the original formats were kept.

Reviewer #2: 

First of all raster resolution of explanatory variables at 10’ or 30’ cannot be acceptable, especially to investigate the distribution of a species that lives in the ponds.

Response: We understand the reviewer’s concern. Our main focus is to understand which environmental factors are most relevant to the ecological niche and distribution of the species. We decided to use layers at the resolutions mentioned above in view of the broad distribution of the species, the availability of some layers at only 30’ resolution (i.e., phosphorus), and the fact that there are no available layers that apply to and characterize all lentic water bodies. In addition, many of the occurrence points available for the species in a large portion of Europe were georeferenced using a grid of 10 km. For this reason, using layers of a finer resolution would not be adequate. Finally, for global-scale climate data, the real spatial resolution of the information available (e.g., weather station data for WorldClim) is actually much more like 30-60’, and all of the detail to which we are accustomed comes from interpolation via reference to digital elevation models. We understand the limitations of our study, and that is why we are focused on discussing which factors are important at a broad scale—clearly, other processes are acting at finer spatial scales, and are not treated in this analysis.

Secondly, the authors try to disentangle some methodological issues in SDM/ENM (e.g. extend of the calibration areas, algorithms) that cannot be addressed using very coarse variables and considering only one species.

Response: We respectfully disagree. We have set out to elucidate the factors that delimit and constrain the ecological niche and geographic distribution of this species on a global extent. We have treated those same methodological issues in many other publications (led by Cobos or by Peterson), such that this paper is a platform not to propose or decide those methodological issues, but rather to reflect on lessons proposed and learned in other studies already completed.

Specific methodological comment:

- Records that were outside of, but closer than ~5’ to the edge of environmental layers were moved to the nearest pixel with information – what? are you expected to find similar environmental condition within 100 km2

Response: We understand your concern. Considering the first law of geography—that near things are more similar than far things—which is to say that climate dimensions show substantial spatial autocorrelation, indeed conditions should be similar, especially along the coast, where these points are located. We also note that, given the coarse resolution of the data layers used, these points falling outside of the raster extent by short distances may be “out” only because of how raster layers are represented (i.e., as square pixels). This is, raster layers do not follow coastal lines properly and gaps in area coverage may exist given the gridded nature of these layers.

- the broad distribution of this species makes it difficult for that method to be applied, why? Maybe is due to the coarse resolution of your approach?

Response: We are very familiar with the method in question, and the reason not to use it is not related to the resolution of our layers. One of the limitations of such a method is that it does not work well when processes are run across large regions where the shape of the planet becomes a factor to calculate distances for dispersal. As our species is distributed across most of the world the limitations of this method constitutes a problem for its application, which is why we did not use it. 

- Modeling algorithms: how did you generate the pseudoabsence? how did you calibrate the models,? have you performed some how the cross validation?

Response: Please see details for pseudo-absences in lines 211-220, model calibration and model evaluation in lines 253-266.

- “We transferred all the models across the area comprising the union of the four calibration areas and compared those models to assess whether patterns of suitability values differed as a result of using distinct variables, calibration areas, and algorithms.” WHY? Is it a methodological or ecological MS? Not clear.

Response: This is an ecological study, but assessing whether choices about environmental variables, calibration areas, and algorithms are important because they can and do affect model outcomes. If one did not assess different modeling choices in those dimensions, one would run the risk of interpreting as biological patterns phenomena that are, in reality, just methodological artifacts. As such, we assess a diversity of methodological choices, and only interpret biologically patterns that are robust to different choices.

- Have you considered the multicollinearity among explanatory variables?

Response: Please see details in lines 225-226, 236.

Please consider to read many paper that suggest how perform the SDM/ENM, how create pseudo-absence, how to create ensemble models by combining different algorithms, how to perform cross validation and so on. Please not use a single species to address methodological question, consider to use virtual species for that. Hereunder some useful paper:

Barbet‐Massin, M., Jiguet, F., Albert, C. H., Thuiller, W. (2012). Selecting pseudo‐absences for species distribution models: how, where and how many?. Methods in ecology and evolution, 3(2), 327-338.

Bucklin, D. N., Basille, M., Benscoter, A. M., Brandt, L. A., Mazzotti, F. J., Romanach, S. S., ... Watling, J. I. (2015). Comparing species distribution models constructed with different subsets of environmental predictors. Diversity and distributions, 21(1), 23-35.

Connor, T., Hull, V., Viña, A., Shortridge, A., Tang, Y., Zhang, J., ... Liu, J. (2018). Effects of grain size and niche breadth on species distribution modeling. Ecography, 41(8), 1270-1282.

Muscarella, R., Galante, P. J., Soley‐Guardia, M., Boria, R. A., Kass, J. M., Uriarte, M., Anderson, R. P. (2014). ENM eval: An R package for conducting spatially independent evaluations and estimating optimal model complexity for Maxent ecological niche models. Methods in ecology and evolution, 5(11), 1198-1205.

VanDerWal, J., Shoo, L. P., Graham, C., Williams, S. E. (2009). Selecting pseudo-absence data for presence-only distribution modeling: how far should you stray from what you know?. Ecological modelling, 220(4), 589-594.

Response: Thank you for the suggestions. We are aware of, and have read, all the references mentioned. We are now citing Barbet‐Massin et al. (2012) in the section in which we explain how GLMs were constructed (see line 217). We also cite Connor et al. (2018) in our discussion (see line 401). While clearly differences exist in terms of methodological choices among labs and among researchers, we believe that the methods used in our study are more current than the ones described in the other references suggested.

Reviewer #3: 

The abstract accurately describes the main objective of the study. It explains how the study was done, including the model organism used, without exceeding methodological details. The most important results are summarized, but their significance has not been sufficiently emphasized, specifically for the variation of the final set of variables selected based on the combination of algorithm, calibration area, and spatial resolution.

Response: We have added text to our abstract to highlight the significance of our findings (see lines 27-31).

The authors provide a careful overview of the challenge of selecting appropriate environmental variables when characterizing species' ecological niches, and what still ought to be done. The objectives of the study are clearly specified, but hypothesis are missing. The lack of clear hypothesis could prevent to really understand the significance and the importance of the study.

Response: We included a hypothesis in our introduction, as suggested (see lines 96-99).

The Materials and Methods section provides enough detail to allow suitably skilled investigators to replicate the main steps of the study. However, the lack of specific information (e.g., method of generating pseudo-absences, method details of GLMs' model calibration, R packages adopted to evaluate GLMs' performance of candidate models) does not allow a full understanding of the code provided. Specific information should be included in detail, citing articles you followed for the choices/methods applied.

Response: We have added the details suggested in multiple parts of our methods. Regarding the method to generate pseudo-absences, we describe that in lines 211-217. Although not many R packages were used to perform GLM-related procedures, we made it clear that we did most of this using base function in R (see lines 265-266). Please see lines 253-261 regarding details in GLM calibration and evaluation (this part includes details for Maxent and GLMs).

In addition, I suggest the authors explain why they adopted that specific ratio of the quantity of presence data to the number of background points/pseudo-absence data to fit models, specifying a reference that supports the choice. This is always a very sore point because according to some authors an inadequate number of background points or pseudo-absences (in this study 20,000 points for 964 occurrences; lines 138 and 190-191) could reduce the accuracy of the models. Model accuracy is generally affected both by this ratio, but also by the method used to generate pseudo-absences (see for example Barbet-Massin et al., 2012).

Response: We added some text to explain why we used such a number of points as background/pseudo-absences, including a reference that justifies our decision (see lines 214-217).

Line 160 - 162: "We performed raster aggregation procedures (average of values) on CEC, OC, and pH to match the resolution of BIO variables, and on BIO and SR variables to match the resolution of variables at 30’. [..]" I do not really understand this sentence. Have you used a BIO variable as a snap raster to ensure all cells were properly aligned, and all rasters have the same cell resolutions? If this is the case, please specify the method applied (e.g., nearest neighbour method etc.), if not rephrase the sentence.

Response: We have added two more sentences to specify the way aggregation was performed (see lines 176-177).

Line 162 - 164: "Although the set of variables representing soil conditions used at 10’ differs from the one at 30’, variable selection analyses will help to identify whether the variables selected differ between the two resolutions. [..]" Does any other author support this??

Response: We rewrote our statement to make it easier to understand and to reflect that we consider that our analyses will allow us to check whether some variables that are present in the two groups (e.g., bioclimatic variables) are selected consistently despite being grouped with distinct variables (see lines 177-180).

Line 171 - 176: "Although a new simulation-based approach has been recently suggested as a reliable tool to estimate calibration areas [59], the broad distribution of this species makes it difficult for that method to be applied. Our chosen calibration areas are therefore reasonable options to calibrate models, considering that such areas should reflect what regions could have been accessible to the species and present relevant environments for comparisons [..]" Does any other author support this? Do you have a specific reference to cite?

Response: Thank you for your questions. We do not have a specific reference for this statement. However, we are very familiar with the methods that use simulations for calibration area delimitation. One of the limitations of such a method is that it does not work well when processes are run across large regions where the shape of the planet becomes a factor to calculate distances for dispersal. As our species is distributed across most of the world the limitations of this method constitutes a problem for its application, which is why we did not use it. The reason why we consider the areas selected as reasonable is that we have used all these options before and they all allow us to include areas close to the ones where the species has been reported. Since close areas are more likely to be accessed, we think model calibration processes were done considering relevant environments (see lines 182-190).

Line 216-217: "The latter consideration assumes that using variables for which the entire spectrum of responses can be characterized makes for better models. [..]" Please cite and provide a reference for this sentence.

Response: There are no references for this particular statement, however this is related to the ideas presented in previous studies about truncated and complete responses, and how non-truncated responses benefit model construction and projection (Chevalier et al. 2021, Owens et al. 2013, Peterson et al. 2011, Thuiller et al. 2004). We have added these references accordingly (see lines 239-241). 

Line 218-219: "Biological relevance of variables was determined based on details about the species' natural history, phenology, and physiology in the literature, and our own experience with populations in the field and controlled environments [..]" No bibliographic reference was cited to justify the selection of biologically relevance variables. Please, integrate a citation

Response: Thank you for your suggestion. We have included references as appropriate in lines 241-243.

Line 260: “see below”. Where? Please, specify.

Response: We have removed this part as it makes reference to one of the latest results we obtained, which is properly referenced in the results section (see lines 349-360).

The results relate to the research question, and the language adopted to express results is clear and concise. The tables and figures are appropriate, but very fragmented in many appendix documents. To explore the results in detail, the reader should open as many as 11 documents (.docx format) and have vector graphic software to open 26 images attached in the .eps format. I strong suggest aggregating appendixes and exporting images in a simple file format (e.g., jpg or .tif).

Response: We included the supporting figures and tables as a separate PDF file for easy reading. However, following the journal requirements the original formats were kept.

Reviewer #4: 

In general, this research the authors very well examine the problem due to the selection of environmental variables for niche ecological modelling. The manuscript is well structured in particular in introduction, methods and discussion, less the results that it requires a large number of information as the results of model evaluation (partial ROC, omission rate, Akaike information criterion for GLM and AICc for Maxent), and also the results of jackknife analysis for Maxent and ANOVA for GLM to measure/explore deviances for each environmental variables selected. 

Response: Thank you for your comments. We added a few details to refer to the criteria that selected models met in terms of partial ROC, omission rates and AIC or AICc (see lines 302-303). We would like to keep our text focused on the sets of variables selected, as this was the main focus of our study. Please see lines 328-339 for a description of jackknife and deviance results. 

Moreover, the results obtained for S. polyrhiza niche models are in line with a large number of previous research with different species targets in which the bioclimatic, solar radiation and soil conditions variables were identified as the most important environmental variables that limit species growth at global scale. Consequently, in the manuscript, I miss a clear paragraph that describes innovative aspects of using these environmental variables in order to produce S. polyrhiza niche models. 

Response: We understand the reviewer’s concern and agree that these types of variables have been used in previous studies to characterize niches and potential distributions of species. We would like to clarify that our goal was not to test new types of variables, but rather we wanted to understand which variables from a multiple and diverse set of predictors could be more relevant to understanding what shapes the niche and distribution of the species (at broad-scale). 

I consider that the use of traditional methods to define the set of environmental variables allowed to achieve same or similar results. Please, provide you with a motivation for this question. 

Response: We respectfully disagree. Traditional methods do not include one of the critical steps in our study (i.e., testing multiple model options in which one of the factors changed is the set of variables). This means that traditionally only one set of variables would have been tested with distinct model parameterizations. Considering that including or excluding certain variables play a big role in model performance the answers that we obtained are certainly different and most likely, more appropriate.

Line 10: Change the corresponding author email with an institutional email.

Response: Thank you for the suggestion. The corresponding author prefers to keep the email as is to allow for long term availability of this means of communication.

Lines 134 – 136: I did not understand this sentence. Are environmental variables not at global scale? Please rewrite this sentence, thank you.

Response: The variables are indeed global, but they are restricted to terrestrial areas. We have added some wording to make it clear (see lines 137-138).

… miss the ecological reasons for choosing these environmental variables (bioclimatic, solar radiation, and soil conditions …)

Paragraph “Environmental variables”: In this paragraph misses a description how these environmental variables were important to S. polyrhiza growth.

Response: We have added some text as a general statement of why these variables are important (see lines 153-157). We also added references in the section “Variable selection process” to guide the readers to the original studies that demonstrated the importance of these types of variables for the species at local level or via experimentation (see lines 157-159). 

Lines 157 – 160: Add a table with the environmental variables used describing their ecological importance specific to S. polyrhiza and the spatial resolution.

Response: We have added a table as a supplementary material and not as part of the main manuscript to avoid redundancy between the text and the table, considering changes that have been made based on request from all reviewers (S3 Table).

… several steps in text require more details and adequate motivations, for example miss ecological reason of dimensions and shapes of calibration areas.

Lines 167 – 171: Please add references that they used similar methods to define calibration area, and also indicated the importance of these methods (buffer, concave-hull polygons, ecoregions, and the intersection of previous three areas) as calibration areas for S. polyrhiza.

Response: A motivation for distances used has been added in lines 186-189. We also added references as requested (see lines 193-194). We consider that with the addition of these references, the explanation of areas used in lines 183-186, and the code provided, the readers should now be able to fully follow the steps we performed.

Lines 198 – 200 and Figure 2: In the main text miss a description of the first step displayed in Figure 2 “Inspection and/or treatment”. Furthermore, please add the same number for each step reported in main text in figure 2 to make the reading of manuscript easier.

Response: We changed our text to account for this step included in figure 2 (see lines 224-225). We decided not to include the numbers from the text in figure 2 as this figure describes the process and intermediate set of decisions to be made. Including the numbers in the figure may result in more confusion. 

Lines 212 – 216: The references of Fig. 3 and Fig. 4 were inverted in main text. After, these two figures (Fig. 3, Fig. 4) showed preliminary results of S. polyrhiza ecological niche model. In my researches, I prefer described these results in the results paragraph. Please move these two figures and add their description in results paragraph.

Response: We have corrected the order of the figures in the text. Considering that the interpretation of the results presented in the figures is actually in the first paragraph of results, we would like to keep the place of these figures as it is to make the methods easier to follow using the examples. We understand that the editorial team of the journal will be the one determining the final position of these figures and we would like to wait to see what they consider more appropriate.

Figures 3, 4, 6, 7: Add in the main text or in the caption of these figures the reasons to display in the main text only the results with 10’ in spatial resolution.

Response: The reasons to display only results for variables at 10’ are: 1) the figure in geographic space is more detailed at this resolution; 2) many of the patterns observed are similar; and 3) to be consistent across the document regarding the results shown. We modified our captions to briefly explain this.

Figures 6 and S11: Add a legend of suitability.

Response: The legend has been added.

The use of GBIF and BIEN underestimate the real spatial extension of S. polyrhiza. Can this problem affect the ecological niche model produced?

Paragraph “Modeling algorithms”: Even if the calibration areas include large portion of territory around S. polyrhiza occurrences, these occurrences derived by GBIF and BIEN dataset that could be underestimated the area of target species. Consequently, background or pseudo-absence points could be false negative. How you considered this question?

Lines 367 – 370: Based on this sentence, I have a question. The S. polyrhiza occurrences used in this research could be not include the total real occurrences, can this problem influenced the results of ecological niche models?

Response: If we understood correctly, the reviewer asks whether the inclusion of occurrence records of the plant that are not known because of lack of sampling could influence the results of ecological niche models? Our short answer is yes, and that is directly related to uncertainty derived from data in any modeling exercise (almost impossible to measure because we do not know the truth). However, the degree of influence of new records depends on how environmentally different they are compared to existent ones. Although no other source of occurrence records of this plant exists and we are not able to answer this question directly, we suspect that the records currently used summarize appropriately the whole set of environmental conditions used by the species. Therefore, the addition of further information may not change our characterizations dramatically.

Regarding our background/pseudo-absence data potentially being false negatives, we are not too concerned about that as Maxent does not assume the background to be absences. GLMs are usually constructed with presences and pseudo-absences (ideally absences), however, weighting our presences and pseudo-absences differently helps to reduce problems related to false negatives. This process makes GLMs to be more similar to Maxent, in the sense that pseudo-absences become something more like a background (see our explanation in lines 218-220).

Lines 403 – 408: Add ecological reasons specific and references to S. polyrhiza and/or other species in order to motivate the inclusion of variables with complication in graphical explorations.

Response: We have added a sentence after this part to talk about the importance of making decisions not only based on statistics but also considering ecological considerations (see lines 428-430).

Lines 411 – 413: The temperature is certainly the main driver to limit the species growth at global scale, please add references.

Response: Thank you for the suggestion. Please see two references specifically related to the importance of temperature for the species in line 433 (the sentence before the one indicated here).

Lines 419 – 421: The relationship between the target species and soil variables were more important in the finest ecological niche model. This describe an higher importance of soil condition at this fine scale, please add references to motivate this feature.

Response: Thank you for the suggestion. We would like to refrain from adding this type of interpretation because solid variables were not the same at the two resolutions. We are concerned that a sentence saying that solid variables may be more important at finer resolutions are not totally supported by our results and can lead to misinterpretation.

---

## [Decision Letter · Decision Letter 1]

18 Apr 2023

Broad-scale factors shaping the ecological niche and geographic distribution of Spirodela polyrhiza

PONE-D-22-28513R1

Dear Dr. Cobos,

We’re pleased to inform you that your manuscript has been judged scientifically suitable for publication and will be formally accepted for publication once it meets all outstanding technical requirements.

Kind regards,

Mirko Di Febbraro

Academic Editor

PLOS ONE

Additional Editor Comments (optional):

Reviewers' comments:

Reviewer's Responses to Questions

**Comments to the Author**

1. If the authors have adequately addressed your comments raised in a previous round of review and you feel that this manuscript is now acceptable for publication, you may indicate that here to bypass the “Comments to the Author” section, enter your conflict of interest statement in the “Confidential to Editor” section, and submit your "Accept" recommendation.

Reviewer #1: All comments have been addressed

Reviewer #4: (No Response)

2. Is the manuscript technically sound, and do the data support the conclusions?

Reviewer #1: Yes

Reviewer #4: Yes

3. Has the statistical analysis been performed appropriately and rigorously? 

Reviewer #1: Yes

Reviewer #4: Yes

4. Have the authors made all data underlying the findings in their manuscript fully available?

Reviewer #1: Yes

Reviewer #4: Yes

5. Is the manuscript presented in an intelligible fashion and written in standard English?

Reviewer #1: Yes

Reviewer #4: Yes

6. Review Comments to the Author

Reviewer #1: The authors addressed my previous concerns regarding autocorrelation bias and other few comments. In my opinion it can be accepted in this form

Reviewer #4: General comments

The paper “Broad-scale factors shaping the ecological niche and geographic distribution of Spirodela polyrhiza” has been considerably improved. In this last version all paragraphs are well structured and exhaustive. The authors have clearly answered my questions and added the information required in the manuscript. Only few details require further clarification before its publication.

Specific comments

Lines 245-246. In the pairwise correlation analysis, which algorithm was used? Please indicate.

Lines 245-249. References to Fig. 3 and Fig. 4 are invert.

Lines 257-259. In the caption of Fig. 3 indicate the icon for combinations of variables with high values of correlation.

Lines 317-319. The results report in S4 and S5 tables do not coincide with the affirmation of this phrase. In S4 table, the Omission rates values are greater than 0.05, and also the AICs values in S4 and S5 tables are of the order of the thousands. Please rewrite this phrase.

7. PLOS authors have the option to publish the peer review history of their article (what does this mean?). If published, this will include your full peer review and any attached files.

Reviewer #1: **Yes: **Mauro Fois

Reviewer #4: No

---

## [Editor Report · Acceptance letter]

24 Apr 2023

PONE-D-22-28513R1 

Broad-scale factors shaping the ecological niche and geographic distribution of *Spirodela polyrhiza*

Dear Dr. Cobos:

I'm pleased to inform you that your manuscript has been deemed suitable for publication in PLOS ONE. Congratulations! Your manuscript is now with our production department. 

Kind regards, 

on behalf of

Dr. Mirko Di Febbraro 

Academic Editor

PLOS ONE